# Spectral variability of gravity-wave kinetic and potential energy at 69°N: a seven-year lidar study

Mohamed Mossad [1], Irina Strelnikova [1], Robin Wing [1], Gerd Baumgarten [1], and Michael Gerding [1]

[1]Leibniz Institute of Atmospheric Physics at the University of Rostock, Kühlungsborn, Germany

**Correspondence:** Mohamed Mossad (mossad@iap-kborn.de)

**Abstract.** We present the first multi-year study of gravity-wave (GW) kinetic $E_{kin}$ and potential $E_{pot}$ energy spectra in the polar middle atmosphere, based on simultaneous temperature and horizontal-wind measurements by the Doppler Rayleigh-Mie-Raman lidar at the ALOMAR observatory (69°N, 16°E). The analysed 7-year dataset (2017-2023) comprises 61 soundings, each longer than 12 h, totalling 2036 h of observations between 35-60 km. Our results reveal a strong winter-summer contrast that depends on both frequency and vertical wavenumber: winter spectra exhibit significantly higher energies and greater variability, particularly at long periods, while summer spectra are marked by a distinct near-inertial peak, most pronounced in $E_{kin}$. We further reveal that the ratio of kinetic to potential energy depends systematically on frequency and vertical wavelength, with near-inertial, short-vertical-scale waves channelling a disproportionate share of energy into kinetic form, indicating that the two-dimensional GW spectrum cannot be treated as fully separable. Frequency spectra also show, for the first time, broken power laws at 35-40 km, merging into a single flatter power law with increasing altitude. These findings provide new constraints on the spectral energy budget at high latitudes in the middle atmosphere and deliver essential benchmarks for validating and improving GW parameterisations in climate and numerical weather prediction models.

**Keywords.** spectra, gravity waves, lidar, temperature, wind, energy, climatology

## 1 Introduction

Gravity waves (GWs) are oscillations in a stably stratified atmosphere which occur as a result of the restoring force of gravity when air is displaced from equilibrium. These waves are generated mainly in the troposphere by different sources, including convection, airflow over mountains, body forcing, and jet streams (Fritts and Alexander, 2003; Crowley and Williams, 1987; Fritts, 1989; Meyer et al., 2018). GWs propagate vertically (upward and downward) and horizontally through the atmosphere, transporting both energy and momentum. When these GWs break, they contribute significantly to the transfer of energy and momentum of the atmosphere, which ultimately affects global circulation (Lindzen, 1981; Holton, 1983; Ern et al., 2018). Direct measurements of the spatial and temporal spectra of GWs are therefore indispensable, since they reveal how GWs actually behave in the real atmosphere with changing altitude and time. In addition, these spectra reveal how the total energy of these waves is partitioned across frequencies and wavenumbers and how their momentum flux is distributed across phase speeds. They can also help constrain and validate GW parameterisations (Alexander et al., 2021).

The superposition of GWs has been observed to give rise to a "universal" or "canonical" spectrum characterized by its logarithmic slope (i.e. power-law exponent) and amplitude (VanZandt, 1982; Dewan et al., 1984; Gardner et al., 1995; Zhang et al., 2006, 2017). Various theories laid out how the form of this GW spectrum might evolve and how wave amplitude could be constrained by different saturation processes (e.g. instability and nonlinear wave-wave interaction) (Dewan and Good, 1986; Smith et al., 1987; Weinstock, 1990; Gardner, 1994) and/or turbulence (Weinstock, 1985; Lindborg, 2006; Pinel and Lovejoy, 2014). We refer to Mossad et al. (2024) Table A1 for a short summary of published observed and theoretical values of the slope of the GW frequency ($\omega$) and wavenumber ($k, l, m$) spectrum, which were estimated from perturbations of different geophysical variables ($T, u, v, w$ and $\rho$). While there is some level of universality of spectral amplitudes and slopes, especially at large vertical wavenumbers $m > m_*$ (where $m_*$ is the characteristic vertical wavenumber which marks the transition between the low-$m$ unsaturated to the high-$m$ saturated GWs) (Allen and Vincent, 1995), models set by saturation theories do not always capture all aspects of observed spectral variability. For instance, theoretical predictions often underestimate spectral amplitudes, the observed kinetic-to-potential energy ratios are typically larger and exhibit greater variability than expected (Nastrom et al., 1997; de la Torre et al., 1999), and spectra of perturbations in strong shears tend to have shallower slopes than those suggested by saturation theories (Eckermann, 1995). The frequency spectrum of temperature and horizontal wind velocity perturbations (while generally uniform) is also observed to be quite variable at different ranges between the inertial frequency $f$ and Brunt-Väisälä frequency $N$ (Chen et al., 2016; Lindgren et al., 2020).

Waves excited from different sources cover different ranges in this spectrum and have distinct frequencies, phase speeds, and horizontal and vertical wavenumbers (Plougonven and Zhang, 2014; Kalisch et al., 2016; Medvedev et al., 2023). As they propagate upward or downward in the atmosphere, they experience refraction, reflection, amplification, and dissipation which result in variation of their spectral properties over time and at different layers of the atmosphere. Climatological studies of GWs observed through temperature, density, pressure, and wind velocity perturbations have yielded valuable insights into their average behaviour, variability, and sources. These studies have also characterized the seasonal, altitudinal, and latitudinal variability of dominant wave scales with increasing altitude (Chu et al., 2018; Ern et al., 2018), estimated the ratio of average kinetic to potential energies (Nastrom et al., 1997), assessed the energy budgets in reanalysis datasets such as MERRA-2 and ERA5 (Podglajen et al., 2020; Strelnikova et al., 2021), and provided statistical estimates of atmospheric instability probabilities (Nozawa et al., 2023). These diagnostic studies are fundamental for tracing how momentum and energy are vertically coupled between the stratosphere and mesosphere.

Characterizing the seasonal and altitudinal variability of GW kinetic and potential energy spectra is crucial for understanding their role in the energy and momentum dynamics of the stratosphere and mesosphere. Previous work on the topic is limited due to the challenging nature of continuously measuring the atmosphere with the high vertical and temporal resolutions required to resolve GWs (tens to hundreds of meters vertically and minutes in time) over altitudes between 30 and 70 km. On the one hand, rocket measurements provide very high-resolution vertical wavenumber spectra (Schöch et al., 2004) but short flight times make them unsuitable for measuring frequency spectra. Additionally, obtaining sufficient rocket data to generate a robust climatology of GW spectra would be cost prohibitive. On the other hand, super-pressure balloon studies can provide important climatologies of quasi-Lagrangian frequency spectra of the kinetic and potential energies in the lower stratosphere (Podglajen

et al., 2016, 2020; Lindgren et al., 2020) but cannot provide information on their corresponding vertical wavenumbers or determine the altitudinal variation. Thus, Rayleigh lidar data are the best option for measuring GW spectra in the middle atmosphere (30 to 70 km). In this altitude range, most previous lidar studies have used temperature and/or density measurements to derive GW vertical wavenumber spectra such as Chanin and Hauchecorne (1981); Whiteway and Carswell (1994); Alexander et al. (2011); Llamedo et al. (2019), frequency spectra such as Gardner et al. (1995); Sica and Russell (1999); Le Pichon et al. (2015); Baumgarten et al. (2018) and horizontal wavenumber spectra by Knobloch et al. (2023). These temperature and density perturbations are typically used to derive the spectra of potential energy in the atmosphere. To access the complementary kinetic energy spectra, a Doppler lidar is needed to simultaneously measure the horizontal wind velocities. There are only a few case studies of wind spectra (vertical or temporal) published from lidar measurements in this altitude range, including Hertzog et al. (2001); Zhao et al. (2016, 2017b); Xue et al. (2020); Strelnikova et al. (2020). A lidar climatology of GW spectra from simultaneous horizontal wind and temperature measurements has not previously been published due to the challenge of measuring winds between 30 and 70 km.

Motivated to address these gaps, we present the first statistical study of the seasonal and altitudinal variations of measured GW kinetic and potential energy spectral amplitudes and slopes derived from Doppler Rayleigh lidar observations of horizontal wind and temperature between 35 and 60 km at the ALOMAR observatory (69°N, 16°E). GW energy spectra in the stratosphere and lower mesosphere can be reliably derived from the ALOMAR lidar measurements as its high-power and daytime capability yield uniquely long datasets (> 12 hours) over a large vertical extent (∼ 25 km) at high temporal and spatial resolution with reasonably low uncertainties. These extended soundings allow us to accurately resolve very low-frequency (long-period) waves in the frequency spectra at different independent altitudes and significantly improve the statistical robustness of vertical wavenumber spectra by averaging spectra of numerous vertical profiles per sounding, thereby providing a more complete characterization of GW spectra. The location of the ALOMAR observatory is also unique: being at the Arctic coast and in close proximity to the Scandinavian mountain ridge, it is ideally positioned to study GWs generated by orography and modified by the land-sea transition, in a region that is often described as the epicentre of climate change (Serreze and Barry, 2011; Hu et al., 2016). Moreover, the daytime capability of the ALOMAR lidar allows continuous measurements of stratospheric and mesospheric GW activity even during polar summer, similar to the Antarctic Fe lidars at the McMurdo Station (Chu et al., 2018; Zhao et al., 2017a) and previously at the Davis Station (Kaifler et al., 2015). However, the ALOMAR Doppler system is unique in that it provides long-term simultaneous temperature and horizontal wind observations using its twin-beam configuration.

The rest of the paper is organised as follows: Sec. 2.1 describes the ALOMAR Doppler Rayleigh lidar and the preprocessing steps applied to the temperature and wind profiles, Sec. 2.2 summarises the seven-year dataset, in Sec. 2.3 we describe the procedure used to extract the GW perturbations. In Sec. 2.4 we introduce the spectral approach to determine the kinetic and potential energy densities of GWs. Two representative winter- and summer-time soundings that illustrate the full analysis chain are discussed in Sec. 3. The climatological results are presented in Sec. 4: observed-frequency spectra in Sec. 4.1.1, altitude-dependent changes in observed-frequency slopes in Sec. 4.1.2, vertical-wavenumber spectra in Sec. 4.3, and the seasonal behaviour of the kinetic-to-potential energy ratio in Sec. 4.4. Finally, in Sec. 5 we present a summary of our relevant

results and conclusions.

## 2 Data and procedure

### 2.1 Lidar description

The Doppler Rayleigh-Mie-Raman (DRMR) lidar in the Arctic Lidar Observatory for Middle Atmosphere Research (ALO-
100 MAR) located in northern Norway (69°N, 16°E), has been measuring atmospheric temperature, stratospheric aerosols, and mesospheric ice particles for 30 years so far (von Zahn et al., 2000; Fiedler and Baumgarten, 2024). The capability for current horizontal wind measurements was reported in 2010 (Baumgarten, 2010), with first wind observations in 2009. The lidar has been used in many previous studies of GWs including climatologies of stratospheric and mesospheric temperature (Schöch et al., 2008), potential energy densities (Fiedler et al., 2004; Strelnikova et al., 2021), GW patterns in noctilucent clouds (Kai-
105 fler et al., 2013; Wilms et al., 2013; Ridder et al., 2017), analysis of inertia GWs (Baumgarten et al., 2015), and variability of the average kinetic and potential energies during January (Hildebrand et al., 2017). This DRMR lidar is unique in being the world's only lidar capable of measuring both winds and temperature in the strato- and mesosphere during daytime. The ability to measure in both daytime and in darkness allows the lidar to measure continuously for tens of hours if the weather is suitable (i.e. clear skies). The lidar set-up consists of two high-power Nd:YAG lasers emitting at 1064 nm, 532 nm, and 355 nm; two
1.8 m tiltable telescopes, and one polychromatic detection system with Raman, Doppler-Rayleigh, and Mie channels (Fiedler and Baumgarten, 2024).

Vertical temperature profiles are calculated along each line of sight using the hydrostatic integration technique (Hauchecorne and Chanin, 1980; Wing et al., 2018). The average temperature, $T$, is then calculated using an uncertainty-weighted mean of temperatures produced along each line of sight. Temperature measurement uncertainties are primarily due to statistical Poisson
noise and are typically 6.5 K at 80 km and less than 1 K at 50 km for the data used in this study. In this study, we use average vertical temperature profiles with a vertical resolution of 150 m and a temporal resolution of 5 minutes. Horizontal wind profiles are measured using an Iodine gas cell and the single edge technique (Baumgarten, 2010). Since the ALOMAR DRMR lidar has two lines-of-sight, we decompose our horizontal wind measurements into vertical profiles of zonal, $u$ and meridional, $v$ wind. Wind profiles have the same resolution as the vertical profiles of temperature and have a random error of $\approx 0.6\,\mathrm{m/s}$ at
50 km and 10 m/s at 80 km (Baumgarten, 2010). For the present analysis, we retain only those altitude and time bins whose random uncertainty lies inside the $1\sigma$ contour of the two-dimensional uncertainty distribution calculated separately for each sounding. The mask obtained in this way is applied identically to both temperature and horizontal-wind soundings, ensuring that all subsequent spectral estimates are based on a common subset of well-resolved data.

## 2.2 Observations

The lidar soundings are constrained by weather conditions and may include measurement gaps. For this analysis, we selected soundings longer than 12 hours after accounting for observational gaps. This criterion allows the estimation of background temperature and horizontal wind velocities based on temporal means. Additionally, a 12-hour GW time series is sufficiently long to sample waves with periods close to the inertial period at ALOMAR ($1/f = 12.8\,\mathrm{h}$). Soundings fulfilling this 12-hour criterion were not further standardized in length, primarily because longer measurements allow us to resolve very low-frequency oscillations (e.g. GWs which are Doppler shifted to frequencies lower than $f$), therefore gaining finer frequency resolution, and providing us with more averaging of vertical wavenumber spectra. Moreover, as our focus is on spectral characteristics rather than mean energy content, large-scale waves like planetary waves and tides can be resolved at their respective frequencies without the need for temporal (high-pass or harmonic) filtering.

A total of 100 soundings were collected over 7 years (2017-2023), of which 61 soundings were used in this analysis, corresponding to 2036 hours, with a mean duration of 33.8 hours per sounding. Tab. 1 lists the total yearly number of soundings, the corresponding total sum of durations in hours, and the sum of analysed sounding durations in summer and winter. Figure 1a shows the distribution of recorded sounding durations which range from a minimum of 12 hours to a maximum of 187 hours with a median sounding length of 19.6 hours. Measurements span the entire year, with a higher frequency of soundings during continuous lidar operation periods in June and July conducted to support noctilucent clouds trend studies, and fewer but longer clear-sky periods in late winter (January and February), see Fig. 1b. We chose to focus on winter and summer differences throughout this study, since they are both opposite extremes in terms of GW activity and background conditions (Strelnikova et al., 2021), and they offer the longest set of observations (20 and 41 soundings, respectively). Here and throughout the paper, winter months are defined as January and February, while summer months are defined as June, July, and August. No soundings from December met the minimum length criterion.

| Year | Number of soundings | Total sum of hours | Sum of hours in summer | Sum of hours in winter |
|------|---------------------|--------------------|------------------------|------------------------|
| 2017 | 12 | 514 | 200 | 206 |
| 2018 | 17 | 543 | 150 | 318 |
| 2019 | 10 | 304 | 139 | 121 |
| 2020 | 14 | 301 | 174 | 33 |
| 2021 | 10 | 308 | 191 | 53 |
| 2022 | 15 | 340 | 176 | 14 |
| 2023 | 22 | 451 | 261 | - |
| **Sum** | 100 | 2760 | 1291 | 745 |

**Table 1.** Yearly number of soundings of $T, u, v$ by the ALOMAR RMR Lidar and their corresponding sum of hours, lasting for more than 12 h. Corresponding sum of hours recorded in summer and winter are also shown.

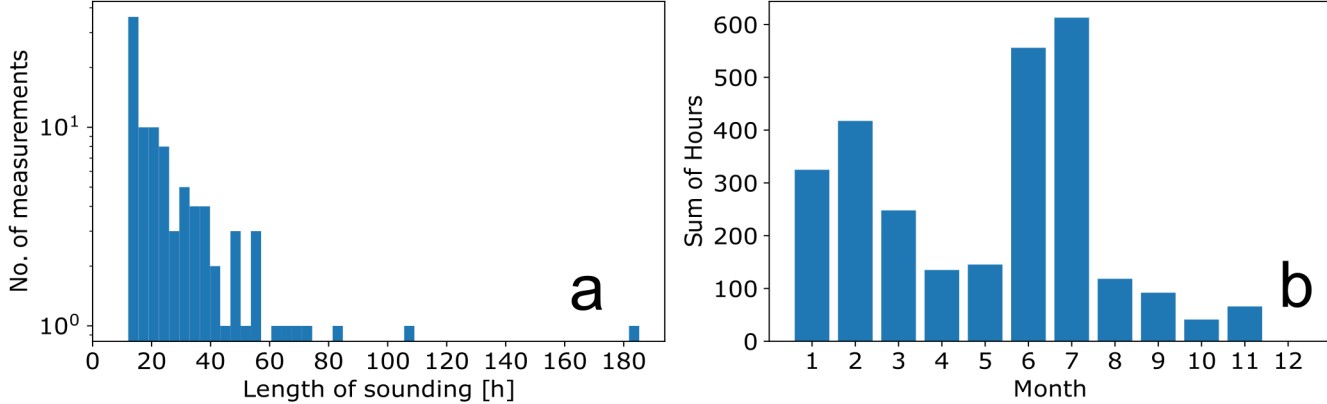

**Figure 1.** Summary of the measurement dataset collected over a 7-year period. Panel (a) shows a histogram of sounding durations, with sounding length on the x-axis (in hours) and the number of measurements on the y-axis, plotted on a logarithmic scale. Panel (b) displays a histogram of the total recorded hours per calendar month, with months (1 = January, ..., 12 = December) on the x-axis and the sum of recorded hours on the y-axis.

## 2.3 Extraction of $T'$, $u'$ and $v'$

Each temperature and horizontal wind sounding ($T$, $u$ and $v$) is assumed to consist of a slowly varying background state $\overline{T}$, $\overline{u}$ and $\overline{v}$, which is taken to be constant over the sounding, in addition to the perturbations $T'$, $u'$ and $v'$. This separation of a
variable into a mean and a deviation from that mean is commonly referred to as a Reynolds decomposition (Holton and Hakim, 2013). These perturbations are mainly due to GWs but also include contributions from tides and planetary waves. We estimate the background from the (uncertainty-weighted) temporal mean of each sounding at each altitude. Then this background is subtracted from the sounding so that the residuals are the perturbations that we seek (Gardner et al., 1989; Ehard et al., 2015). By subtracting the mean, stationary waves are removed and very low-frequency waves remain dominant across the spectrum
(Chu et al., 2018). As the last step applied, a $30\,\mathrm{km}$ (uncertainty-weighted) running mean is subtracted from the perturbations at each time step. While this step limits the vertical wavenumber spectrum at the longest scale, it is necessary; as it rectifies the problem of long vertical stripes in Rayleigh temperature soundings (Zhao et al., 2017a).

## 2.4 Spectral procedure

We present spectra of the temperature and horizontal wind perturbations in the following kinetic and potential energy density normalization to enable physically meaningful interpretation and comparison. This normalization places both variables on the same scale ($\mathrm{J/kg}$ or $\mathrm{m^2 s^{-2}}$), allowing direct assessment of their relative contributions to the total GW energy. It also facilitates calculation of the kinetic-to-potential energy ratio, a key diagnostic for wave dynamics. Moreover, expressing spectra in energy terms allows for consistent comparison with theoretical models and supports climatological analyses of wave energy. The

spectra of potential and horizontal kinetic energy densities $E_{pot}$ and $E_{kin}$ are defined as functions of vertical wavenumber $m$ and observed frequency $\omega$ by:

$$E_{pot}(m,\omega) = \frac{1}{2}\text{PSD}\left(\frac{g(z)}{N(z)}\frac{T'(z,t)}{\overline{T}(z)}\right), \tag{1}$$

$$E_{kin}(m,\omega) = \frac{1}{2}\left(\text{PSD}\left(u'(z,t)\right) + \text{PSD}\left(v'(z,t)\right)\right), \tag{2}$$

where $g$ is the gravitational acceleration, $N$ is the Brunt-Väisälä frequency which is calculated from the estimated $\overline{T}$ (Wing et al., 2021), and PSD is the power spectral density (Nastrom et al., 1997; Tsuda et al., 2000). Here, no additional squaring of $T'/\overline{T}$, $u'$, or $v'$ is required because the PSD is already defined as the squared modulus of the Fourier-transformed signal. We estimate the one-dimensional PSD using the uncertainty-weighted Generalized Lomb-Scargle (GLS) (Zechmeister and Kürster, 2009), which performs better where measurement noise is present than Fourier-based approaches. Before estimating the PSD, the perturbations of the normalized relative temperature and the horizontal wind velocities are prewhitened with a first order autoregressive fit. Their spectra are subsequently postdarkened to compensate for prewhitening (Chen et al., 2016). Prewhitening the signal and postdarkening its spectrum is an essential step to reduce spectral leakage, particularly in the case of very steep spectra (slope $< -2$) and because of contributions from unresolvable scales longer than the duration of a time series or the length of a vertical profile (Dewan and Grossbard, 2000; Mossad et al., 2024). Note that all spectra reported in this paper are demonstrated in terms of observed periods $\tau = 1/\omega$ and vertical wavelengths $\lambda_z = 1/m$, as this notation is more intuitive.

For frequency spectra, a single spectrum is computed as the PSD of the time series at each altitude separately, and an average spectrum is calculated from the mean of all single spectra over 5 km range to reduce the variance in the spectrum. Analogously, for vertical wavenumber spectra, a single spectrum is computed from vertical profiles at each time step separately, and an average spectrum is calculated from the mean of all single spectra over the whole sounding duration. Since time series at different altitudes in a sounding can vary in length (which determines the largest resolvable scale and frequency spacing), an average spectrum in this paper is calculated as follows: all single spectra are first log-scaled then binned to a common frequency grid, subsequently, the mean spectrum is calculated at each frequency bin and finally rescaled. The binning step takes place by linearly interpolating each individual log-scaled spectrum to a common grid whose bins are defined between the lowest and highest resolvable frequencies in the spectrum whose lowest frequency is the lowest of them all (i.e. the longest sounding or time series). For significance purposes, if the number of resolvable spectral amplitudes at each frequency bin is less than the third of the total number of spectra averaged, this bin is skipped. This procedure is used to average single frequency and vertical wavenumber spectra within each sounding, and also to attain the seasonal average from spectra of different soundings.

Assuming additive white noise, an estimate of the noise floor is subtracted from the resulting average spectra (Wilson et al., 1991; Whiteway and Carswell, 1995), which should offer a comparable result to efficient de-noising techniques like the interleaved method (Jandreau and Chu, 2024). Subtracting the estimated noise floor also helps reveal physically meaningful energy

densities, especially in high-frequency regimes where measurement noise can become comparable to the true atmospheric variance. To calculate the spectral noise floor, we use a maximum likelihood fit of a power-law function and a constant term to account for the noise level (Vaughan, 2010). Frequencies (or wavenumbers) higher than and equal to the lowest frequency whose corresponding amplitude is negative (i.e. less than the estimated noise floor) are ignored and not shown after the subtraction. Thus, noise-subtracted spectra from different soundings will extend to different high frequencies or wavenumbers due to different noise levels.

## 3   Example results

To show an example of the analysis to our data, two soundings of $T'$, $u'$ and $v'$ data recorded by the ALOMAR RMR lidar in summer 2021 (1-4 June) and winter 2018 (30 January-4 February) are selected and shown in Fig. 2 after being smoothed with a Gaussian kernel using a full width at half maximum of $1\,\mathrm{h}$ and $1\,\mathrm{km}$, merely for better visual clarity. However, no smoothing is done to the data or the spectra at any step throughout the analysis. Notwithstanding small temporal gaps, the summer and winter soundings are $85\,\mathrm{h}$ and $105\,\mathrm{h}$ long, respectively. These soundings are long enough to reveal distinct atmospheric circulation patterns across time and altitude, which significantly differ between the two seasons.

Because the perturbations have near-zero mean by construction, we quantify their typical magnitude with the root-mean-square (RMS) amplitude. In the winter case (30 January-4 February 2018; left column of Fig. 2), the lidar-derived temperature perturbations reach an amplitude of $\mathrm{RMS}_{T'} \approx 5.9\,\mathrm{K}$, whereas the horizontal-wind perturbations attain $\mathrm{RMS}_{u'} \approx 16.6\,\mathrm{m\,s^{-1}}$ and $\mathrm{RMS}_{v'} \approx 15.2\,\mathrm{m\,s^{-1}}$ throughout the altitude range 35-60 km. The spectral distribution of the perturbations is broad, displaying multiple superposed wave packets of diverse vertical wavelengths and observed periods that together resemble a "wave soup" of large and small scale waves.

By contrast, the summer case (1–4 June 2021; right column of Fig. 2) exhibits markedly weaker activity: the corresponding RMS amplitude of the perturbations is $\mathrm{RMS}_{T'} \approx 2.1\,\mathrm{K}$, $\mathrm{RMS}_{u'} \approx 9.8\,\mathrm{m\,s^{-1}}$ and $\mathrm{RMS}_{v'} \approx 8.9\,\mathrm{m\,s^{-1}}$. These low-amplitude perturbations in summer are persistently dominated by quasi-monochromatic waves with nearly parallel phase lines and apparent periods close to the inertial period of $12.8\,\mathrm{h}$. Such narrowband behaviour is consistent with the lower-stratospheric radar results of Nastrom and Eaton (2006), which showed that more than half of their summer wind measurements show quasi-monochromatic near-inertial oscillations dominating with increasing height, whereas they were rarely observed in winter. Nevertheless, the ALOMAR case study by Baumgarten et al. (2015) showed that similar inertia GW signatures can be observed in winter too. Planetary waves are also more commonly observed in winter perturbations than in summer, especially near the stratopause (Chandran et al., 2013). These differences are typical of the two seasons at ALOMAR and are not special features of the presented cases.

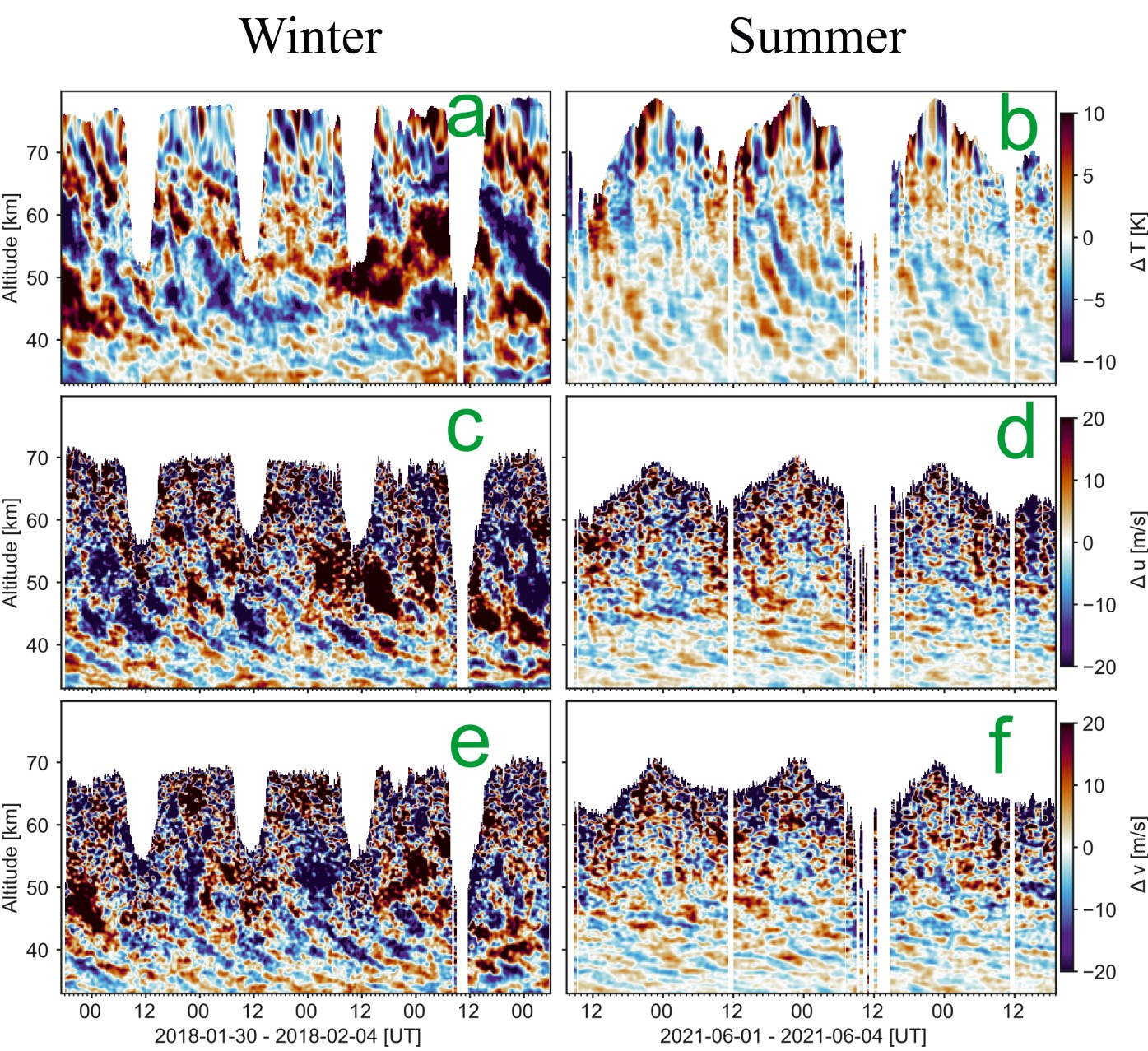

**Figure 2.** Temperature (a)-(b), zonal wind (c)-(d), and meridional wind (e)-(f) perturbations from example soundings selected from the ALOMAR lidar 7-year dataset. The left panel column corresponds to the winter case (30 January to 4 February 2018), and the right column to the summer case (1 June to 4 June 2021). Data are smoothed using a Gaussian window with a width of 1 h in time and 1 km in altitude, merely for improved visual clarity. Altitude (33–80 km) is plotted on the vertical axis, and universal time (UT) on the horizontal axis. Data gaps are plotted as blank.

## 3.1 Observed frequency spectra

To place the characteristics of the perturbations in a spectral context, we now examine the observed frequency spectra of potential $E_{pot}$ and kinetic $E_{kin}$ energy densities derived from the same summer and winter soundings introduced in Fig. 2. Figure 3a and Fig. 3b present these spectra averaged over the 35-40 km range. With a vertical resolution of 150 m, this interval contains about 34 individual spectra, which are averaged to reduce variance and get a smooth estimate of the spectral shape. Each panel displays the spectra before and after subtraction of the estimated noise floor. These spectra show that the winter case exhibits larger potential and kinetic energy densities than the summer case across all frequencies, except for a pronounced peak at 9.2 h in summer, where both energies rise to winter-like levels. This 9.2 h feature exceeds the 95% confidence level and another way to visualize it is if we were to extrapolate a line from high to lower frequencies, it would lie entirely beneath the observed peak. To translate these spectral features into average energy densities per unit mass, we integrate each spectrum—for consistency—over the common 12.8-3.1 h band. The resulting kinetic energies are $5.2\,\mathrm{m^2\,s^{-2}}$ in summer and $23\,\mathrm{m^2\,s^{-2}}$ in winter; while the correspondingly integrated potential energies were smaller overall, with $2.2\,\mathrm{m^2\,s^{-2}}$ in summer and $3.5\,\mathrm{m^2s^{-2}}$ in winter. The summer values agree quite well with the reported values from Lagrangian (intrinsic) frequency spectra in the lower stratosphere by Schoeberl et al. (2017).

We next focus on the slope of the frequency spectra, which is an important characteristic of GWs. In the linear theory of GWs, the spectrum of upward-propagating GW energy per unit mass is often assumed to be separable in vertical wavenumber $m$, intrinsic frequency $\hat{\omega}$, and azimuthal propagation direction $\phi$ (VanZandt, 1985; Fritts and VanZandt, 1993). Within this framework, the intrinsic frequency component of the spectrum, $B(\hat{\omega})$, typically follows a power-law form, $B(\hat{\omega}) \propto \hat{\omega}^{-p}$, where the slope $p$ is commonly taken to be around $5/3$ between $f \ll \hat{\omega} \ll N$. This reference slope is plotted as a dashed $-5/3$ line in Fig. 3; however, both theoretical and observational studies suggest that frequency-spectrum slopes can span a broader range (Fritts and Alexander, 2003; Mossad et al., 2024). This is also the case where Fig. 3 shows that the observed spectra do not exactly coincide with the $-5/3$ reference. A closer look at Fig. 3a and Fig. 3b reveals that the frequency spectrum of $E_{kin}$ steepens toward lower frequencies and develops a broad peak centred near $f$, whereas $E_{pot}$ is well described by a single, nearly constant slope across the analysed band and can therefore be modelled with a simple power-law form. Nevertheless, we fit the standard power law function $B(\omega) = B_0\,\omega^{-p}$ to compute the spectral slope for both $E_{kin}$ and $E_{pot}$, where $B_0$ is a normalization factor and $\omega$ is the observed frequency. This means the reported slope of the log-scaled spectrum is equal to $-p$. Because the slope drastically changes near (but not exactly at) the inertial frequency $f$, the fit is applied at frequencies higher than the transition frequency $\omega^*$ (Chen et al., 2016). The kinetic and potential energy frequency spectra transition to a positive slope at 11.5 h and 10.1 h in summer, and a quasi-flat slope at 10 h and 13 h in winter, respectively. At higher frequencies, the fits from Figure 3 show that $E_{pot}$ has a steeper slope of $-2.11 \pm 0.18$ in the summer case than the winter case's $-1.81 \pm 0.13$. $E_{kin}$ also shows a similar pattern, where the fitted slope for the summer case is $-2.51 \pm 0.61$, while in winter, it is slightly shallower at $-2.28 \pm 0.34$. These very steep slopes and large uncertainties stem from $E_{kin}$ steepening near $f$, which can be accounted for by modifying the power law to a form similar to $\omega^{-p}\left(1 + \frac{f^2}{\omega^2}\right)$ (Hertzog et al., 2002). These results indicate comparatively less energy in the summer case at short periods but a relative enhancement at the near-inertial-scale to amplitudes comparable

to the winter case.

## 3.2 Vertical wavenumber spectra

Analogous to observed frequency spectra, we show the average vertical wavenumber spectra of the two cases in Fig. 3c and Fig. 3d, which represent the mean of spectra of all vertical profiles between 35-45 km in each sounding. Having a temporal resolution of 5 min and quite long soundings, the spectra of the winter and summer cases are averages of 1221 and 895 individual spectra, respectively. This is why the vertical wavenumber spectrum of a sounding is much smoother than its frequency spectrum, and no distinct dominant scales can be seen. Here also, both energy densities of the winter example are greater than those

of the summer example up to high wavenumbers ($m > 1/2\,\mathrm{cycle/km}$ for $E_{kin}$ and $m > 1\,\mathrm{cycle/km}$ for $E_{pot}$) where $E_{kin}$ and $E_{pot}$ amplitudes are comparable in both cases. Whether this feature, along the comparable energy densities between summer and winter cases at long observed periods ($\sim 9\,\mathrm{h}$), is associated with the same wave components is examined in Sec. 4.4 using the ratio of $E_{kin}$ to $E_{pot}$ (from seasonal average spectra) as a diagnostic. The integrated $E_{kin}$ between 10-1.5 km had a value of $7.4\,\mathrm{m^2 s^{-2}}$ in summer and $38\,\mathrm{m^2 s^{-2}}$ in winter, while the integrated $E_{pot}$ were $1.9\,\mathrm{m^2 s^{-2}}$ in summer and $5.9\,\mathrm{m^2 s^{-2}}$ in winter.

Another distinction between $E_{kin}$ and $E_{pot}$ is manifested in their spectral shapes. While Fig. 3c shows that $E_{pot}$ can be described by a simple power law, Fig. 3d shows that $E_{kin}$ scale differently in different wavenumber regimes, as in the observed frequency spectra discussed above. To quantify the spectral shape, we use a bending power-law model $A(m) = A_0 \frac{(m/m_*)^s}{1+(m/m_*)^{s+t}}$ with a pre-defined slope $s = 0$ at low wavenumbers $m < m_*$, to compute the slope $t$ at high wavenumbers ($m > m_*$) (Allen and Vincent, 1995). We estimate that $E_{kin}$ drastically changes slope at 1.7 km in the summer case and 3.7 km in the winter

case. At shorter wavelengths, $E_{kin}$ bends, and shows corresponding slopes of $-4.5 \pm 0.65$ in the summer case and $-3.3 \pm 0.35$ in the winter case. At longer wavelengths ($m < m_*$), the slopes of $E_{kin}$ correspond to $-0.7 \pm 0.8$ in the summer case and $-1 \pm 1$ in the winter case. In contrast to $E_{kin}$, $E_{pot}$ exhibit more subtle transition at 4.3 km in the winter case, showing slopes at $m > m_*$ corresponding to $-2.2 \pm 0.22$ in the summer case and $-2.9 \pm 0.05$ in the winter case.

To put these slope and amplitude values into perspective, and in line with the previously described separable GW theoretical

spectrum, a reference model drawn from the linear instability theory (LIT) is shown in Fig. 3c and Fig. 3d (Dewan and Good, 1986; Smith et al., 1987). It is given for $E_{pot}$ and $E_{kin}$ by:

$$E_{pot}^{LIT} = d\frac{N^2}{2(2\pi)^2 m_*^3}\frac{1}{1+\mu^3} \tag{3a}$$

$$E_{kin}^{LIT} = b\frac{N^2}{2(2\pi)^2 m_*^3}\frac{1}{1+\mu^3} \tag{3b}$$

where $\mu = m/m_*$, $m_*$ is the characteristic vertical wavenumber, $d \approx 1/10$ and $b \approx p \cdot d \approx 1/6$ with $p$ being the slope of intrin-

290 sic frequency spectra (Fritts et al., 1988; Nastrom et al., 1997). The factors 10 and 6 in the denominators of $E_{pot}^{LIT}$ and $E_{kin}^{LIT}$ were empirically inferred from observations and the prediction that the ratio of $E_{kin}(m)/E_{pot}(m)$ is equal to $p \sim 5/3$ which should remain constant over the wavenumber spectrum (VanZandt, 1985; Nastrom et al., 1997). Nevertheless, these values can

be tuned to be consistent with observations (Gardner, 1996). In this context, saturation theories (like the LIT) constrain the amplitude of GW vertical wavenumber spectrum (at $m > m_*$) by different saturation mechanisms, with limits proportional to $N^2$ within a factor of two of the expressions given by the LIT above (Fritts and Alexander, 2003). This high wavenumber region ($m > m_*$) is typically located at vertical scales of approximately a few kilometres in the stratosphere. This $m_*$ characterizes the transition from the saturated GW regime with a slope of $-3$ towards less steep spectral slopes at lower vertical wavenumbers $m < m_*$ in the unsaturated GW regime, which is dominated by source characteristics (Allen and Vincent, 1995). The LIT model shown in Fig. 3 (with $m_* = 1/4\,\mathrm{cycle/km}$) is consistent with the observed $E_{pot}$ amplitude and shape in the winter case, but it only matches the shape of $E_{pot}$ in the summer case, whose amplitude is constantly smaller by a factor of $\sim 5$. The observed $E_{kin}$, on the other hand, exceeds the LIT model amplitudes in winter and shows a similar shape, whereas it flattens early in summer and falls below the LIT model values at long wavelengths. Perhaps, a better agreement in amplitude for observed $E_{kin}$ in the winter case with the LIT model can be achieved by adjusting the value of $b$ in Eq. 3b to be equal to $1/2$. This discrepancy is likely due to the fact that the LIT model assumes a constant ratio of $E_{kin}/E_{pot}$, which is not always the case in observations (de la Torre et al., 1999). Furthermore, Fritts et al. (1988) discussed that if the amplitudes of low intrinsic frequency waves are limited by dynamical instabilities, then the factor $b$ in Eq. 3b can be modified to $\left[\frac{2}{1+(1-f^2/\hat{\omega}^2)^{1/2}}\right]^2$. This means that if a near-inertial GW dominates the frequency spectrum, then the saturated $E_{kin}^{LIT}$ amplitudes given in Eq. 3b should exhibit significant increase and by extension the ratio of $E_{kin}/E_{pot}$ at the corresponding wavenumbers.

## 4 Climatological results

Building upon the analysis of the two selected cases of individual soundings presented in the previous section, this section presents the climatological analysis of GW energy density spectra derived from the seven-year dataset described in 2.2 at altitudes between 35 and 60 km. The seasonal and altitudinal variability in observed frequency and vertical wavenumber spectra of both kinetic ($E_{kin}$) and potential ($E_{pot}$) energies are thoroughly examined. Particular attention is given to how these spectra conform to theoretical expectations, vary between summer and winter, and evolve with altitude. Additionally, we explore the variability of slopes of frequency spectra and the kinetic-to-potential energy ratio $E_{kin}/E_{pot}$, which provide further diagnostic insights into wave characteristics and the underlying seasonal modulation of GW dynamics.

### 4.1 Observed frequency $\omega$-spectra

Figure 4 presents observed frequency spectra of the kinetic ($E_{kin}$) and potential ($E_{pot}$) energy densities derived from the temperature and horizontal-wind perturbations in winter and summer. Each spectrum line represents the average of all single spectra in the 35 to 40 km layer in a sounding, after the noise floor of each sounding's average spectrum is subtracted. For comparison, spectra of the previously selected cases (01 June 2021 and 30 January 2018) are also shown alongside soundings from their respective seasons. The spectra of both cases seem to conform to the seasonal mean's spectral shape, although the summer case's long-period (>6 h) energy densities and winter case's energy densities across the whole spectrum lie on the

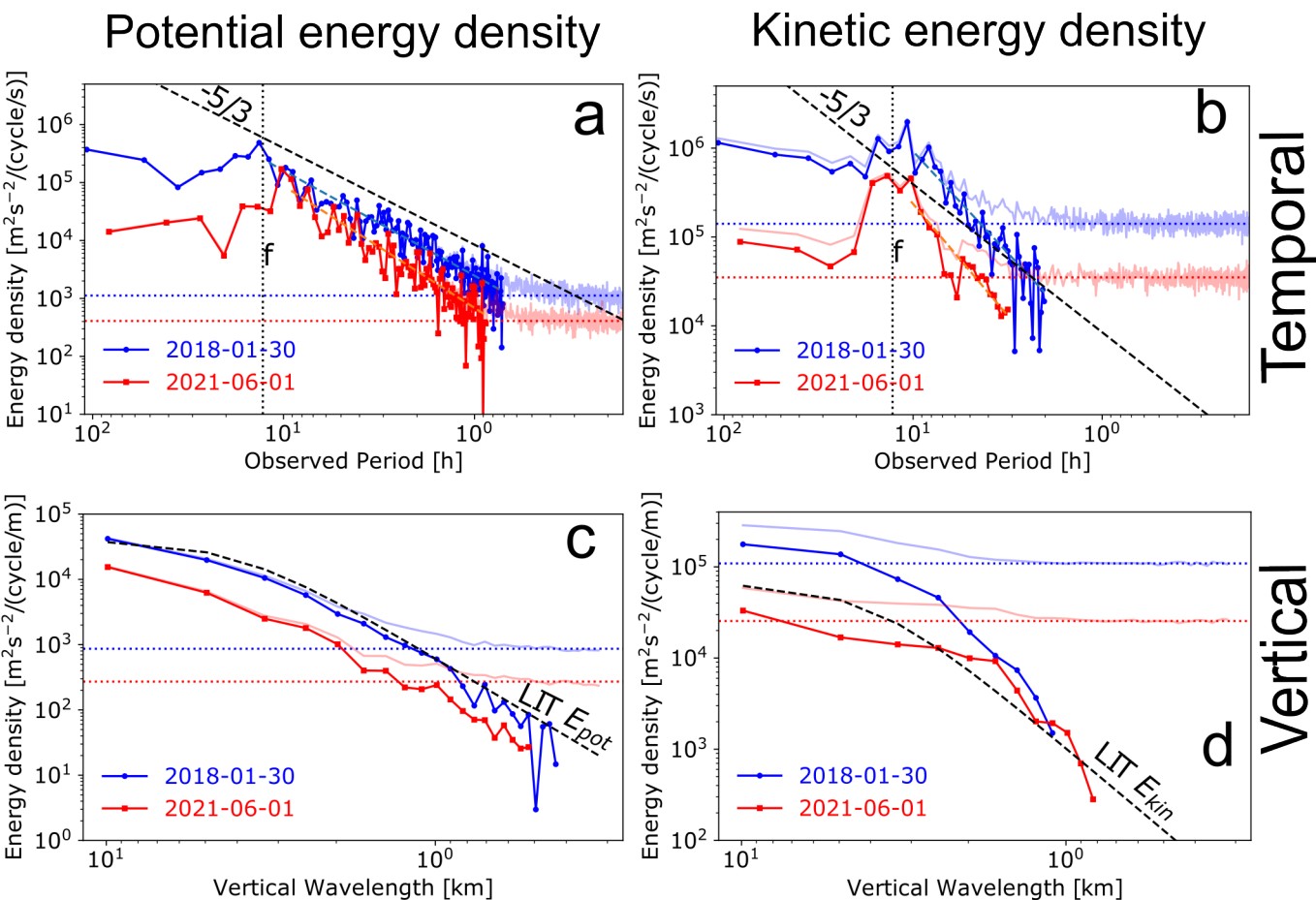

**Figure 3.** Energy density spectra of the summer and winter cases, shown in Fig. 2 before (transparent) and after (opaque) subtraction of their noise floors. Panels (a) and (b) display the potential and kinetic energy density spectra, respectively, as a function of observed period (in hours), calculated from the average of spectra over the 35-40 km altitude range. Panels (c) and (d) show the corresponding potential and kinetic energy density spectra as a function of vertical wavelength (in kilometres), calculated from vertical profiles between 35-45 km and averaged across time. Subtracted noise levels are shown as dotted lines, with colours corresponding to their respective seasonal spectrum. Energy density is given in units of $m^2 s^{-2}$ per cycle per unit frequency or wavenumber (cycles per second or per meter). Black dashed lines indicate reference slopes of -5/3 (frequency spectra), and -3 (vertical wavenumber spectra, LIT).

higher end of their distributions. The seasonal mean shown in Figure 4 lies in the range of $10^3 - 10^6\,\mathrm{m^2 s^{-2}}/(\mathrm{cycle/s})$ for both $E_{kin}$ and $E_{pot}$ at intermediate frequencies (observed periods between 2-8 h). At periods longer than the inertial period (12.8 h), the spectra generally either flatten (winter) or significantly drop (summer), whereas toward higher frequencies (shorter periods) they typically decline steeply. This $\omega > f$ behaviour is broadly consistent with GW intrinsic frequency spectra, which often exhibit a slope near $-5/3$ as mentioned above (Fritts and Alexander, 2003). Notably, spectra of each season's soundings show

considerable variability in amplitude from each other, reflecting the intermittent nature of GWs. However, aside from changes in amplitude and shifts in spectral slope near $f$, their overall seasonal mean remains relatively stable and there is an apparent

conformity in the spectral shape. This can also be seen as the upper and lower quartiles seem to generally follow the seasonal mean at each frequency bin.

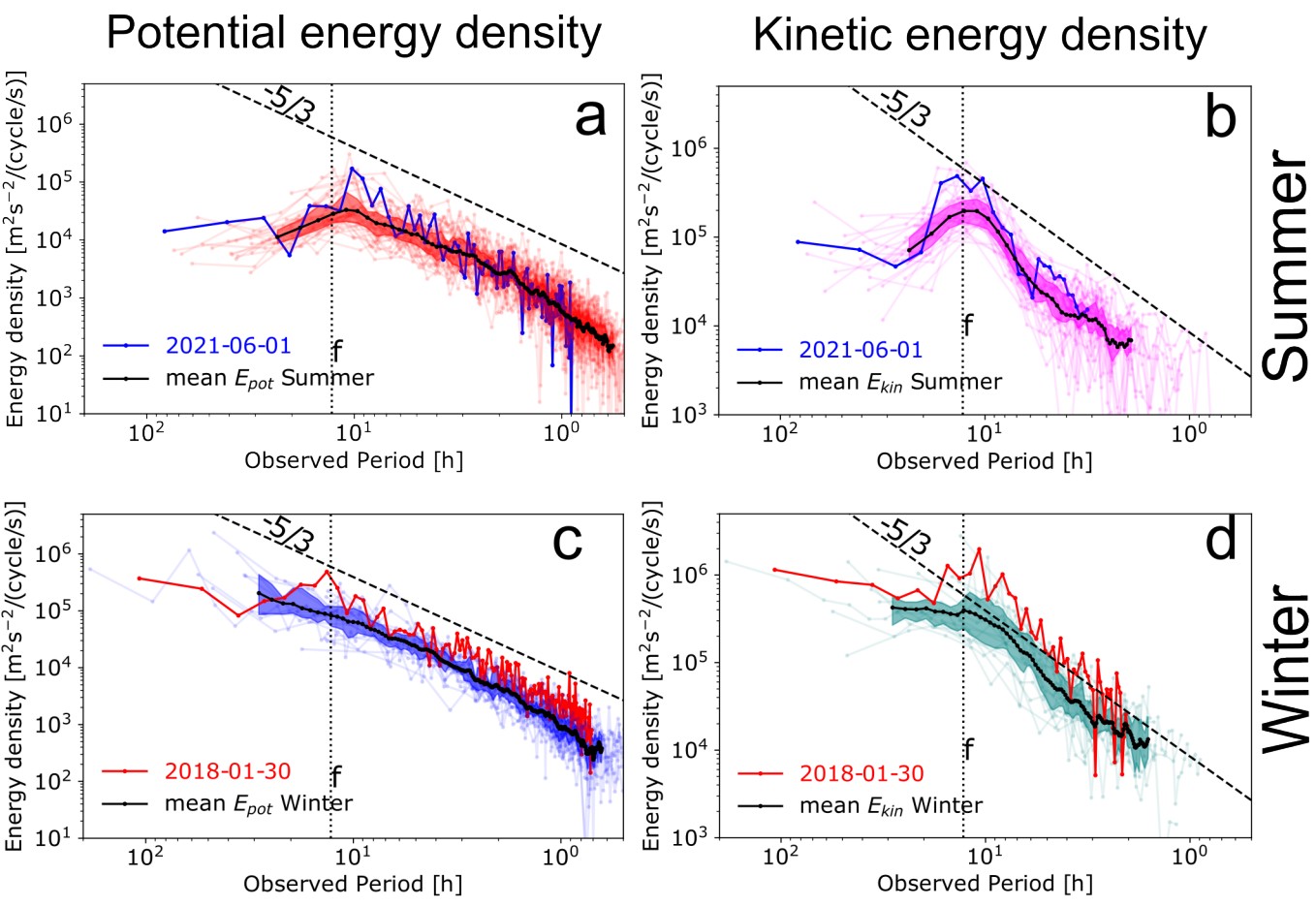

**Figure 4.** Temporal energy spectra of potential (a and c) and kinetic (b and d) energy densities for all data recorded in Tab. 1 plotted on a log-log scale. Each spectral curve corresponds to the average of all temporal spectra between 35 and 40 km of a sounding. Superimposed on these individual spectra are the seasonal mean spectra (black solid lines). Solid lines in red and blue belong to the spectra of the summer and winter cases shown in Fig. 3. The Coriolis frequency at ALOMAR $f \approx 1/12.8\,\mathrm{h}^{-1}$ is shown as a dotted black vertical line, and a reference slope $= -5/3$ (dashed black line) is shown. The colour shading represents the upper and lower quartiles (25 and 75 percentiles) in each frequency bin.

### 4.1.1 Seasonal and altitudinal variation of $\omega$-spectra:

Having discussed the overall spectral behaviour at 35-40 km, we now explore how the average energy-density spectra vary with season and altitude. Figure 5 shows the seasonal averages of $E_{kin}$ and $E_{pot}$, obtained by averaging all individual soundings from winter and summer. Shaded bands denote the corresponding interquartile ranges (IQR), computed in the same way as in

Fig. 4. Two altitude layers are first considered: the mid stratosphere (35-40 km, Fig. 5a) and the lower mesosphere (55-60 km, Fig. 5b). As an estimate of energy content, see Tab. 2 for the mean integrals of the spectra of each season's soundings between $12.8 - 4.1$ h and their standard deviations. Between 35-40 km (Fig. 5a), the winter spectra generally exhibit higher overall amplitudes than their summer counterparts, especially at low to intermediate frequencies ($1/10$ cycle/h $< \omega < 1/1.5$ cycle/h). This result is consistent with enhanced GW activity in winter, often driven by stronger background winds, more prevalent wave sources (jets and fronts) in the polar stratosphere (Strelnikova et al., 2021), and GWs interacting with larger scale waves. These typical winter conditions allow more efficient vertical propagation of large-amplitude waves. In contrast, the amplitude of summer spectra remains lower across all frequencies, consistent with enhanced background filtering of GWs due to wind reversal in the lower stratosphere (Wilson et al., 1991).

This seasonal contrast in Fig. 5a is evident for observed frequencies near and lower than $f$ as well. In particular, a local minimum exclusively emerges in the amplitudes of summer spectra (both $E_{pot}$ and $E_{kin}$) at periods longer than 10 h, whereas winter spectra exhibit either flattening ($E_{kin}$) or a continuous increase ($E_{pot}$) in this synoptic-scale regime. This minimum corresponds to the spectral gap between gravity and Rossby waves previously noted in intrinsic frequency spectra of quasi-Lagrangian measurements by Hertzog et al. (2002) and Podglajen et al. (2020), however, this gap is visible only in our summer observations and is absent in winter. Within this same long-period range -specifically the 10-30 h band- only GWs and tides can contribute to our spectra. Because tidal amplitudes at these heights are small in temperature and similarly low in winds relative to long period GWs (Baumgarten et al., 2018; Hagen et al., 2020; Baumgarten and Stober, 2019), with no dominant harmonic peaks; the excess winter energy below $f$ is most plausibly attributed to strongly Doppler-shifted GWs. This effect is illustrated in Fig. 1 of Gardner et al. (1993), which showed that as the mean wind velocity approaches or exceeds the intrinsic horizontal phase speed of the dominant GW, progressively more energy is Doppler shifted to frequencies lower than $f$ and higher than $N$. This mechanism can explain why this part of the spectrum is seasonally distinct.

Notably, we also observe a broad local maximum just above the inertial frequency $f$, which is particularly more prominently pronounced in $E_{kin}$, see Fig. 5a. It is broader (between frequencies $f < \omega < 1/4$ cycle/h) in winter spectra than in summer, where it is sharply peaked (between frequencies $f < \omega < 1/6$ cycle/h). Although we do not conduct spectral rotary analysis here to investigate the nature of such waves, the observed peak near $f$ in $E_{kin}$ aligns qualitatively with the inertial GW signatures reported by Hertzog et al. (2002); Gelinas et al. (2012); Conway et al. (2019); Podglajen et al. (2020). This energy surplus near $f$ suggests vertical wavenumbers near $m_*$ (Fritts and Alexander, 2003), where near-inertial GWs tend to have small vertical wavelengths (Hertzog et al., 2002). The seasonal modulation of this peak is consistent with Doppler shifting by the background wind: in winter, stronger and more variable stratospheric winds (Strelnikova et al., 2021) can distribute a larger band of near-inertial intrinsic GWs across (and even outside) the observed spectrum, broadening the enhancement, whereas the weaker summer winds confine the shift to a narrower range, yielding a sharper peak. The fact that the feature is much more prominent in $E_{kin}$ than in $E_{pot}$ further supports an interpretation in terms of near-inertial motions, whose polarisation favours higher kinetic than potential energy. The relation of $E_{kin}$ to $E_{pot}$ and the corresponding vertical scales of these near-inertial GWs are further examined in Sec. 4.4.

By 55-60 km altitude (Fig. 5b), namely, the lower mesosphere, both energy spectra follow a broadly similar shape as in the

|  | Winter $E_{kin}$ | Summer $E_{kin}$ | Winter $E_{pot}$ | Summer $E_{pot}$ |
|---|---|---|---|---|
| **35-40** km |  |  |  |  |
| Energy integral between 12.8-4.1 h $[\mathrm{m^2s^{-2}}]$ | $5.8 \pm 4.9$ | $2.4 \pm 1.2$ | $1.4 \pm 1$ | $0.8 \pm 0.48$ |
| slope $p$ [for $\omega > f$] | $1.83 \pm 0.17$ | $1.94 \pm 0.29$ | $2.03 \pm 0.09$ | $1.99 \pm 0.13$ |
| **40-45** km |  |  |  |  |
| Energy integral between 12.8-4.1 h $[\mathrm{m^2s^{-2}}]$ | $10.5 \pm 8.3$ | $4 \pm 2.2$ | $3 \pm 2.2$ | $1.3 \pm 0.6$ |
| slope $p$ [for $\omega > f$] | $1.93 \pm 0.25$ | $2.04 \pm 0.37$ | $2.01 \pm 0.08$ | $1.83 \pm 0.13$ |
| **45-50** km |  |  |  |  |
| Energy integral between 12.8-4.1 h $[\mathrm{m^2s^{-2}}]$ | $25 \pm 23.4$ | $6.1 \pm 3.6$ | $6.5 \pm 6$ | $2.2 \pm 1.5$ |
| slope $p$ [for $\omega > f$] | $1.85 \pm 0.24$ | $1.65 \pm 0.56$ | $1.91 \pm 0.09$ | $1.73 \pm 0.16$ |
| **50-55** km |  |  |  |  |
| Energy integral between 12.8-4.1 h $[\mathrm{m^2s^{-2}}]$ | $29.8 \pm 18$ | $7 \pm 3$ | $7.8 \pm 6.1$ | $3.1 \pm 2.3$ |
| slope $p$ [for $\omega > f$] | $2.06 \pm 0.29$ | $1.88 \pm 0.62$ | $1.97 \pm 0.11$ | $1.44 \pm 0.16$ |
| **55-60** km |  |  |  |  |
| Energy integral between 12.8-4.1 h $[\mathrm{m^2s^{-2}}]$ | $37.8 \pm 29.6$ | $16.7 \pm 13.4$ | $12.1 \pm 9.6$ | $4.4 \pm 3.4$ |
| slope $p$ [for $\omega > f$] | $1.73 \pm 0.4$ | $1.38 \pm 1$ | $1.84 \pm 0.12$ | $1.45 \pm 0.23$ |

**Table 2.** Table of total potential and kinetic energies per unit mass in $\mathrm{m^2s^{-2}}$, estimated by integrating the spectra over periods between 12.8 and 4.1,h and averaging these integrals across all individual spectra. Also shown are the slopes of the mean frequency spectra for $\omega > f$. The uncertainties for the energy integrals represent the standard deviation of the mean values, reflecting the variability across all spectra. The uncertainties for the slopes correspond to the fitting errors from the power-law regression.

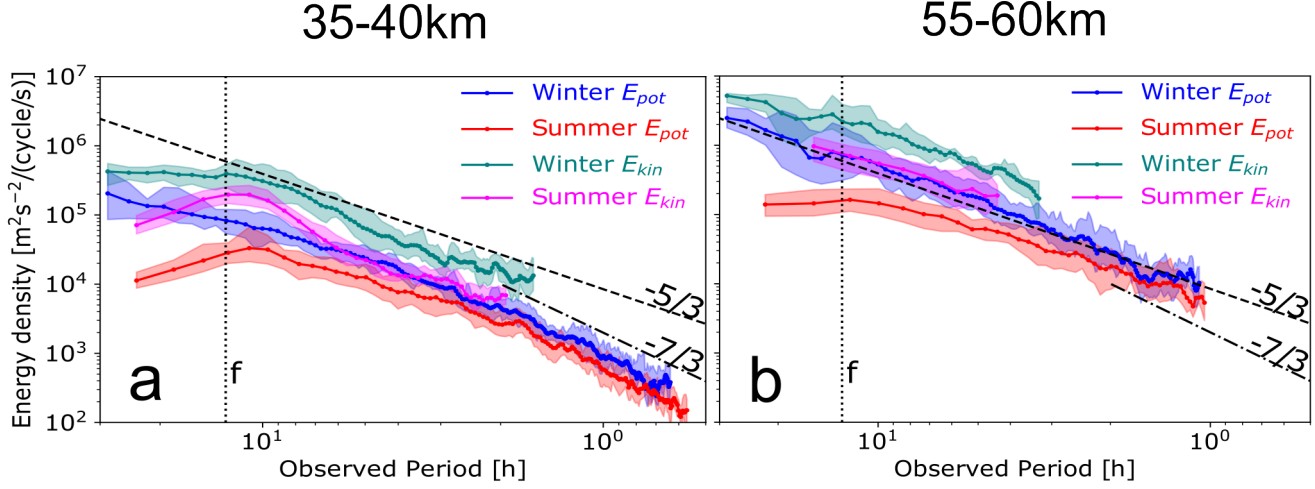

**Figure 5.** Mean temporal spectra, showing the seasonal and altitudinal variability of both $E_{kin}$ and $E_{pot}$.

stratosphere. The relation between winter and summer spectral amplitudes of both energy densities also remains quite similar
to lower altitudes, see Fig. 6 for 5 km layer comparison. Except for the layer 50-55 km, which shows a peak in the winter-to-
summer ratio at intermediate frequencies, no pronounced altitudinal variation is seen in either $E_{pot}$ or $E_{kin}$. At frequencies
$\omega < f$, the ratio of winter-to-summer amplitudes reaches a maximum value of 25 for $E_{pot}$ and 10 for $E_{kin}$, then exponentially
decreases as the frequency increases towards $f$. At frequencies $\omega > f$, winter spectral amplitudes remain greater than summer

at longer periods ($> 4\,\mathrm{h}$) and the ratio fluctuates at $2.5 \pm 0.5$ for $E_{pot}$ and at $3 \pm 1$ for $E_{kin}$. However, at shorter periods than

$3\,\mathrm{h}$, the $E_{pot}$ winter-to-summer ratio fluctuates at only $1.5 \pm 0.5$. This indicates that low- and intermediate-frequency waves propagate more effectively during winter up-to and throughout the 35-60 km range without extensive dissipation compared to summer.

The vertical evolution seen by comparing the mean spectra in 35-40 km up to 55-60 km further highlights how waves from different frequency bands are preferentially attenuated or amplified with altitude. For instance, the distinct inertial peak near $f$

observed in $E_{kin}$ spectra at 35-40 km is more broadened and basically disappears by 55-60 km. This may imply that as near-inertial waves ascend, they either dissipate or are Doppler-shifted to periods outside the resolved spectral range. In contrast, high-frequency components (i.e. periods $< 2\,\mathrm{h}$) grow more rapidly than lower frequency waves with altitude. To illustrate this pattern, an additional steeper line with a slope of $-7/3$ is included in Fig. 5 as a reference at periods smaller than $2\,\mathrm{h}$. This $-7/3$ line matches the spectral shape of $E_{pot}$ in the high-frequency regime of Fig. 5a which is steeper than the lower frequency

part of the spectrum whose slope is closer to $-5/3$ or more positive. As a result, we can identify power-law breaks in the 35-40 km spectra of both seasons at $\approx 2\,\mathrm{h}$. This seasonal similarity for periods smaller than $3\,\mathrm{h}$ is not manifested in the spectral slopes only, but also winter amplitudes are only about 50 % larger than summer, as seen in Fig. 6.

These breaks are less identifiable in winter since the slope of the lower frequency regime is steeper ($\sim -1.7$) than in the summer spectra ($\sim -1.4$), see Fig. 7. Figure 5b shows that the 55-60 km spectra behave differently. In this height layer, even

though the $E_{pot}$ spectra are only resolved up to $\omega \approx 1/0.9\,\mathrm{cycle/h}$, it is clear that as altitude increases the broken power law merges into a simple power law. Slopes of this power law are shifted to more positive values as altitude increases, especially in summer. This indicates that if the entire spectral range is considered for fitting a single power-law model, these observed breaks and peaks would bias the resulting slope estimates, leading to mistakenly steepened spectral slopes in summer (Nastrom and Eaton, 2006). Thus, caution should be exercised in interpreting slopes of the whole spectrum, and segmented spectral

fitting might better represent the underlying GW dynamics, see Tab. 2 for the determined slopes of available average spectra (for $\omega > f$) and Sec. 4.1.2 for extensive analysis of range-dependent slopes. This pattern is only quantifiable in $E_{pot}$ because temperature perturbations are less noisy, which allow the resolved $E_{pot}$ to extend to very high frequencies, even up to $1\,\mathrm{h}$ at 60 km. Nevertheless, $E_{pot}$ relates to $E_{kin}$ through the GW polarization relations and should have comparable slopes except near $f$ and $N$ (Schoeberl et al., 2017).

The reason the high-frequency band in the 35-40 km spectra has a steeper slope can be traced to selective removal of low-frequency waves in summer, produced by critical-level filtering. At ALOMAR, low stratospheric winds reverse from strongly eastward in winter to weaker westward in summer (Strelnikova et al., 2021). When a GW with an observed frequency $\omega$ and horizontal wavenumber $k_h$ encounters a background flow $\bar{U}(z)$ such that its intrinsic frequency $\hat{\omega} = \omega - k_h\bar{U}$ tends to zero, its vertical wavelength shortens and the wave is absorbed or breaks. Although the magnitude of summer winds is smaller, the

sign reversal ensures that slow westward-propagating waves meet a critical level below $\sim 30\,\mathrm{km}$ and are removed from the spectrum (Whiteway and Duck, 1996; Sato and Yoshiki, 2008). The result is a high-pass filtered spectrum in which the relative contribution of high-frequency components ($\omega \gtrsim 1/2\,\mathrm{cycle/h}$) increases because the low-frequency part is attenuated. With increasing altitude, the low-frequency deficit persists (the filtered waves cannot re-enter), while the surviving high-$\omega$ waves are

amplified as background density decreases with altitude. This could explain how the broken power law gradually merges into a single power law by 55-60 km. Additionally, strong wave breaking above $\sim$40 km can spawn secondary GWs that may populate the high-$\omega$ end of the spectrum. In winter, however, eastward mean winds allow the entire frequency band to propagate upward without strong critical-level filtering so that the slope of the average spectrum increases by only $\approx 0.19$ from 35-40 km up to 55-60 km, whereas it increases by $\approx 0.54$ in summer, see Tab. 2.

This slope variation with height is accompanied by a systematic growth of mean energy density. To obtain an estimate of this growth, we integrated energy spectra from different height layers between 12.8-4.1 h for consistency; see Tab. 2. From 35-40 km up to 55-60 km, the kinetic energy $E_{kin}$ increases in both seasons on average by a factor of 1.7 with every 5 km step, whereas the potential energy $E_{pot}$ rises by a factor of about 1.8 per step in winter and 1.5 per step in summer. This amplification is consistent with wave-action conservation, which predicts upward propagating GW amplitudes to grow as $\sim e^{z/2H}$ (where $H$ is the scale height) and hence energy growth $\propto \exp(z/H)$ as background density decreases exponentially with height $z$ (Fritts and Alexander, 2003). With a representative scale height of 7-10 km, theory predicts layer-to-layer factors near 1.6-2.0; our observed range of 1.5-1.8 thus falls well within the anticipated range. The slightly greater growth of $E_{pot}$ in winter reflects the stronger background winds, which favour efficient upward transport of large-amplitude GWs.

Another important distinction between summer and winter is the variability across the spectrum. As a measure of spectral variability, we determine the standard deviation (SD) of energy density (integrated between 12.8-4.1 h) which expands markedly with increasing height and is always larger in winter than in summer: From 35-40 km up to 55-60 km, the SD of $E_{kin}$ broadens from 1.2 to 13.4 in summer and from 4.9 to 29.6 in winter, whereas that of $E_{pot}$ expands from 0.48 to 3.4 in summer and from 1 to 9.6 in winter. See Tab. 2 for SD values in 5 km steps. As a dynamical cause for winter high variability, the strong polar-night jet and associated vortex serve both as a vigorous source of large-amplitude waves and as a selective waveguide (Wing et al., 2025). This behaviour is documented in satellite momentum-flux climatologies and modelling studies (Ern et al., 2016). Thus, as altitude increases, the spectrum not only gains energy but also spreads more, with variability amplifying in step with the mean.

### 4.1.2 $\omega$-spectra slope variability

To further investigate how each part of all observed frequency spectra changes with season and altitude, we determine the slope changes in $E_{pot}(\omega)$ as a function of altitude layers, between 35-60 km in steps of 5 km. Understanding slope behaviors is a key diagnostic for understanding wave dynamics, saturation processes, and seasonal variability. In addition, slopes derived along different spectral ranges could help extrapolate GW amplitudes to very short periods in (re)analysis datasets (Lindgren et al., 2020; Podglajen et al., 2020) and compare with high-resolution models. We thus use a simple power-law fit to estimate the slope over the full spectral range ($f < \omega < 1/0.5$ cycle/h) as well as in two frequency sub-ranges: intermediate-to-low $1/12.8$ cycle/h $< \omega < 1/2$ cycle/h and high frequency $1/2$ cycle/h $< \omega < 1/0.5$ cycle/h. This fit is only performed if the frequency range to be fitted contains more than five frequency points to ensure statistical significance. We show the average slopes from spectra of all soundings in terms of their median and their IQR for all three ranges in Figure 7, as the slopes are not always

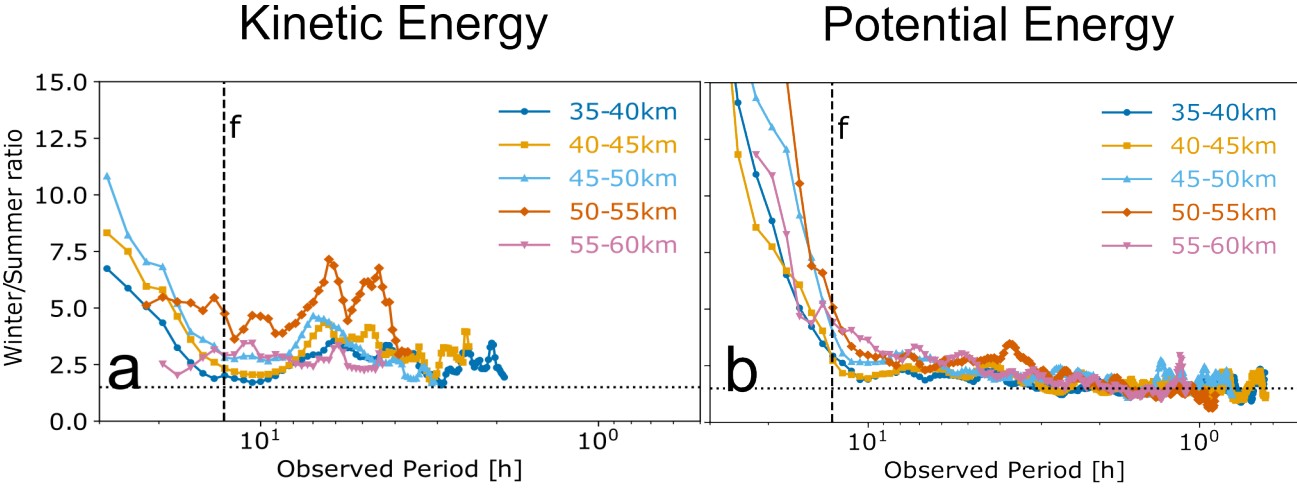

**Figure 6.** Altitudinal variability of the ratio of mean winter to summer amplitudes at different frequencies for both kinetic (a) and potential (b) energy spectra. Dotted line corresponds to the ratio 1.5.

normally distributed. In general, significant differences emerge between the summer and winter slopes at different altitudes. In the range $f < \omega < 1/0.5\,\mathrm{cycle/h}$ which covers nearly the whole spectrum, Fig. 7a shows that as altitude increases up to the
50-55 km layer, the more slopes of both seasons drift apart. Winter spectra are consistently steeper over all altitude ranges, with a quasi-constant median slope of $\approx$-2(-2.1 to -1.9) at 35-40 km and $\approx$-1.8(-2 to -1.7) at 55-60 km. In contrast, summer spectra progressively flatten with altitude, with a median slope of $\approx$-2(-2.1 to -1.7) at 35-40 km, and up to $\approx$-1.5(-1.7 to -1.3) at 55-60 km, i.e. a rate of increase of the slope by $\sim 0.1$ per 5 km.

To understand how these altitudinal shifts of the full-spectral-range slope come to be, we analyse the variability of the two
comprising frequency sub-ranges. Over intermediate and long periods (2.0-12.8 h in Fig. 7c), slopes of the spectra from both seasons show similar altitudinal variation to one another, although winter slopes remain more negative throughout the 35-60 km range. This seasonal difference in slope magnitudes can be attributed to low-frequency waves being filtered in the summer lower stratosphere (as discussed earlier in Sec. 4.1.1), so that the power law does not rise on average with altitude as in winter, where these large-amplitude low-frequency waves propagate upwards more freely. The altitudinal variation is manifested as a slight
increase in steepness in the stratosphere (<50 km) then the opposite happens in the mesosphere (>50 km) where this spectral region becomes flatter. The steepness of average winter slopes slightly increases from -1.7(-2 to -1.4) to -1.9(-2 to -1.5) at 50 km then decreases back to -1.7(-1.9 to -1.6) at 60 km. Similarly, summer slopes increase from -1.6(-1.8 to -1) to -1.7(-2 to -1.1) at 50 km and then decrease to -1.4(-1.8 to -1) at 60 km. This steepening observed below 50 km could be driven by enhanced energy density at frequencies near $f$ (12.8-6 h in Fig. 5a), whereas the flattening above that height arises because
this enhanced feature disappears and energy grows preferentially at higher frequencies (6-2 h in Fig. 5b). Nevertheless, the steepening below 50 km and the subsequent flattening lead to small changes in the slope's magnitude ($\sim 0.2$ and well within the IQR). Therefore, we interpret them as minor fluctuations about an approximately constant average slope rather than as a systematic trend, and therefore avoid over-stating their dynamical significance.

In the high-frequency range (0.5-2.0 h in Fig. 7b), another notable seasonal similarity persists. For instance, the average slope at 35-40 km is approximately -2(-2.2 to -1.9) for both seasons. As altitude increases, the slopes shift to more positive values, reaching values of -1.6(-1.8 to -0.7) and -1.7(-2 to -1.4) at 55-60 km for winter and summer, respectively. Comparing these slopes to the intermediate and low frequency range (2.0-12.8 h in Fig. 7c), a clear distinction is seen: slopes along the high-frequency part of the spectrum are significantly steeper and less variable at altitudes lower than 50 km. This reinforces the statistical significance of the broken power-law observed in mean $E_{pot}$ spectra in Fig. 5. Furthermore, the strong flattening in this range in both seasons indicates that energy at the highest frequencies increases with increasing height. As mentioned earlier, decreasing background density with height and wave breaking above ∼40 km may lead to an increase in energy in this high-frequency region. While this trend is quite similar in both seasons, especially below 50 km, only summer slopes along the full spectral range (0.5-12.8 h) match this strong flattening trend. We thus attribute the winter quasi-constancy of slopes in the full spectral range (0.5-12.8 h) to the slopes in the intermediate and long periods (2.0-12.8 h) being steep enough to counteract this trend.

These seasonal differences in spectral slopes can be understood by considering wave-mean flow interactions, which reinforce the findings from the mean energy spectra themselves. In summer, winds are generally weaker and reversed relative to winter (Wilson et al., 1991), creating critical levels in the lower stratosphere. These critical levels prevent many waves—especially slower-moving, longer-period (low-frequency) modes—from reaching higher altitudes (Wilson et al., 1990). Consequently, at all altitudes, the amplitude of waves in the low- and intermediate-frequency bands in summer is significantly smaller (see Fig. 6), yielding a flatter spectral slope compared to winter. In winter, stronger winds allow broader propagation of these lower-frequency waves upwards, maintaining their amplitude increase and producing a relatively constant, steep slope. This significant seasonal discrepancy at low frequencies in the estimated slopes aligns well with MST radar observations at 32°N by Nastrom and Eaton (2006). In contrast, higher-frequency waves (shorter periods) can more easily avoid critical-level filtering due to their faster phase speeds and shorter vertical scales, showing comparable amplitude and slope in both seasons.

Regarding the slope variability: there is no reason why the slope of an observed-frequency spectrum should remain constant from sounding to sounding, even though each is an average of spectra from 5 km range. For instance, Schoeberl et al. (2017) showed that the distributions of their estimated slopes from Lagrangian $E_{pot}$ and $E_{kin}$ are quite identical and both range between -1.3 and -2.5 with an average of -1.9. The presence of a dominant peak, or flattening of the spectrum at a different transition frequency (e.g. near $f$) can also bias the computed slope (Chen et al., 2016); to a greater extent when a linear least-squares fit is used rather than a maximum likelihood power-law fit (Mossad et al., 2024). This can be seen in Fig. 3a, where the enhanced energy in the spectrum of the summer case near 8-9 h skewed its slope to be steeper than the winter case. Furthermore, it is not expected for observed frequency spectra to behave exactly like intrinsic frequency spectra due to Doppler shifting by the background wind. Still, our spectra are showing consistent behaviour within each season and the full-spectral-range slopes from both seasons are about as variable as intrinsic frequency spectra by Lindgren et al. (2020). This is because high intrinsic-frequency waves, having smaller amplitudes, contribute minimally to altering the overall spectral shape when shifted by Doppler effects, and could only make the spectra less smooth by creating small local peaks and dips as seen in the

mean spectra in Fig. 5. Dominant low-frequency oscillations, on the other hand, are Doppler shifted to observed frequencies which are not very far from their intrinsic frequencies (Fritts and Alexander, 2003). Therefore, the computed slope should not differ substantially from the intrinsic slope if the transition near $f$ is accounted for. The higher variability observed in slopes for the frequency sub-ranges, as opposed to the full 0.5-12.8 h range, likely arises because fewer points are available for fitting, which broadens the distribution of slopes.

Analogous to our results in the high-frequency subrange in Fig. 7b, lidar measurements at 78°S showed progressive flattening of the slope with altitude ranging from -2.6 (in the $\sim$ 1-10 h band) around 85-100 km altitude to around -1.6 above 100 km, suggesting increased energy at high-frequencies with altitude (Chen et al., 2016). Such altitude-dependent flattening of spectral slopes is also reported in the mid stratosphere, where slopes rise from -1.88 at 32.5 km to -1.45 at 62.5 km (Zhao et al., 2017a). This behaviour is often attributed to filtering and dissipation of low-frequency energy with increasing height. Season and latitude can similarly introduce slope variability; for instance, Lindgren et al. (2020) lower stratospheric results showed that June-August spectra between 40°N-North-Pole tend to display steeper slopes relative to December-February, and the opposite happens between South-Pole-40°S. These northern hemispheric results clearly contrast with our findings, as we observe steeper slopes (which are statistically significant) in winter compared to summer. However, Lindgren et al. (2020) noted that potential sampling biases may affect their results. As far as the authors are aware, our study is the first to report a statistically significant broken power law in observed-frequency spectra, particularly evident in summer, where filtering processes differentially affect longer-period waves and lead to a pronounced change in spectral slope. In general, current climatological studies across the 20-110 km altitude range reveal distinct nuances such as excess energy in specific period bands, altitude-dependent slope changes, and seasonal variations.

## 4.2  Vertical wavenumber spectra

Analogous to the frequency spectra, we also analysed the wavenumber $m$-spectra of vertical profiles of temperature and horizontal velocity perturbations from the lidar soundings previously described in Sec. 2.2. Figure 8 illustrates the corresponding $E_{kin}$ and $E_{pot}$, averaged for each sounding in the upper stratosphere (35-45 km), together with their seasonal means and IQR. As with $\omega$-spectra, $m$-spectra of both winter and summer cases generally conform to the shape of the seasonal means. However, $E_{pot}$ amplitudes of the winter case exceed the upper quartile range, while those of the summer case remain within it. For $E_{kin}$, amplitudes of both cases are also larger than the seasonal upper quartiles between between 1.5–3 km in summer and 2.5–10 km in winter. Across low and intermediate wavenumbers (wavelengths between 2–10 km) both seasonal means fall within $0.7 \times 10^3 - 10^5 \, \mathrm{m^2 s^{-2}}/(\mathrm{cycle/m})$ for $E_{kin}$ and $E_{pot}$. Despite some deviations, there is a common shape observed in $E_{kin}$ of both seasons: a steep slope at short wavelengths ($< 3$ km) that transitions into a flatter slope at longer wavelengths. In contrast, $E_{pot}$ follow a simple power law with a slope similar to the high wavenumber region ($m > 1/2 \, \mathrm{cycle/km}$) in $E_{kin}$. See Sec. 4.4 for a detailed comparison between the different amplitudes of $E_{kin}$ and $E_{pot}$.

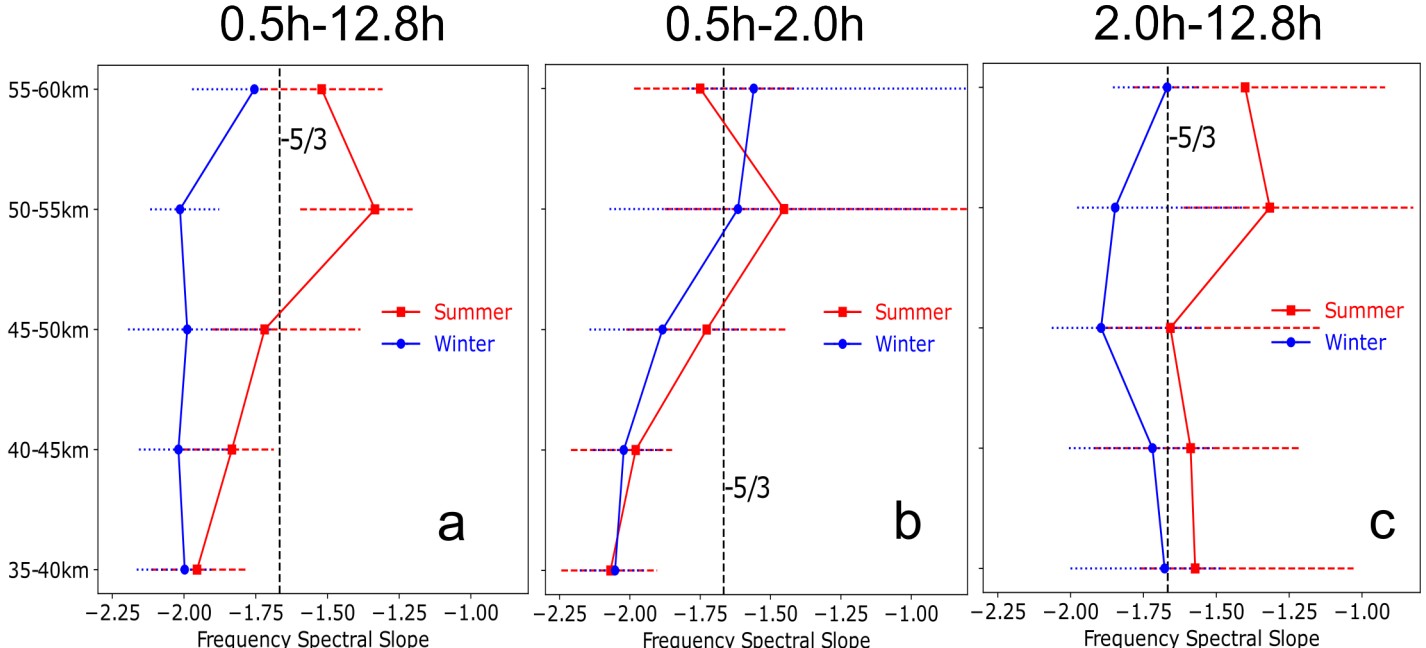

**Figure 7.** Variability of slopes ($-p$) of the observed $E_{pot}$ spectra derived from seven years of ALOMAR lidar soundings, as a function of season and altitude. The slopes are fitted separately in three bands: (a) full observed spectral range (0.5-12.8 h), (b) the short-period range (0.5-2.0 h), and (c) the intermediate-long-period range (2.0-12.8 h). Median slopes from both seasons are represented by blue circles in winter and red squares in summer. Bars mark the interquartile range (IQR).

### 4.3 Seasonal and altitudinal variation of vertical wavenumber spectra:

To further explore how GW energy varies with season and altitude, we compare $E_{kin}(m)$ and $E_{pot}(m)$ from different atmospheric layers. Figure 9 shows the average vertical wavenumber spectra of $E_{kin}$ and $E_{pot}$ in winter and summer from vertical profiles between 35-45 km and 45-55 km. We opted to split the vertical profiles into these two 10 km layers, so that a fair comparison can be assessed between the upper stratosphere and the lower mesosphere. To obtain an estimate of GW energies, we integrated $E_{pot}$ and $E_{kin}$ of each sounding in the wavelengths range of 10-2 km for all layers and listed the mean integrals along their standard deviations in Tab. 3. Figure 9a shows spectra from profiles between 35-45 km where the mean winter $E_{pot}$ and its IQR exhibit greater energy than summer spectra across all wavenumbers. In contrast, summer $E_{kin}$ amplitudes rise to amplitudes comparable to those in winter at short wavelengths ($\lambda_z < 2\,\text{km}$). For $E_{pot}$, however, the ratio of winter-to-summer amplitudes remains fairly constant at 2.5 across the wavenumber range, rather than decreasing with increasing wavenumber. This high-wavenumber result closely resembles the pattern observed in the winter-to-summer ratio of frequency spectra in Fig. 6 at long observed periods (low frequencies) between $13 - 8\,\text{h}$.

Spectra from the 45-55 km layer in Fig. 9b show that there is a noticeable increase in the amplitude of both energy densities in both seasons, compared to the 35-45 km layer. The ratio of the amplitudes in the upper to the lower layer is equal to $\approx 1.8$ between vertical wavenumbers $1/3\,\text{cycle/km} < m < 1/0.8\,\text{cycle/km}$. This ratio is greater at lower wavenumbers reaching

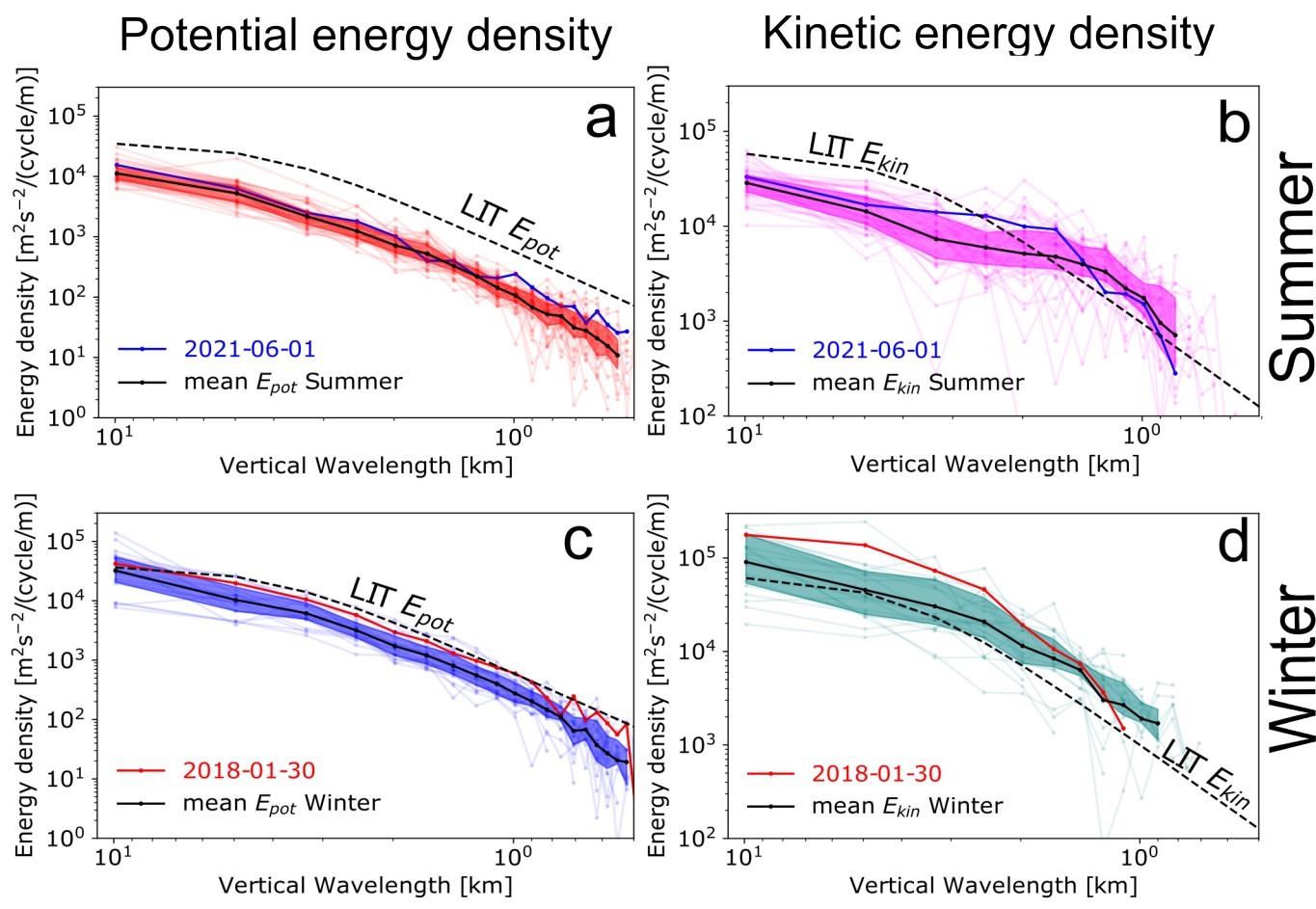

**Figure 8.** Vertical wavenumber energy spectra of potential (a and c) and kinetic (b and d) energy densities for all data described in Tab. 1. Each spectral curve (transparent) corresponds to the mean (over time) of all vertical wavenumber spectra between 35 and 65 km for each sounding. Superimposed on these individual spectra are the seasonal mean spectra (black, opaque lines). Solid lines in red and blue belong to the spectra of the summer and winter cases shown in Fig. 3. The dashed black lines are amplitude prediction by the linear instability theory in Eq. 3 for $m_* = 1/4\,\text{cycle/km}$. The mean values estimated for $N^2$ are $48 \times 10^{-5}\,\text{s}^{-2}$ in winter and $45 \times 10^{-5}\,\text{s}^{-2}$ in summer. The colour shading represents the upper and lower quartiles (25 and 75 percentiles) in each frequency bin.

| | Winter $E_{kin}$ | Summer $E_{kin}$ | Winter $E_{pot}$ | Summer $E_{pot}$ |
|---|---|---|---|---|
| **35-45** km | | | | |
| Energy integral between 10 and 2 km. $[\mathrm{m^2 s^{-2}}]$ | $16 \pm 10.8$ | $4.3 \pm 1.4$ | $3.8 \pm 1.8$ | $1.4 \pm 0.5$ |
| slope $[m > m_*]$ | $-2.8 \pm 0.2$ | $-1.2 \pm 0.4$ | $-3.1 \pm 0.07$ | $-2.4 \pm 0.09$ |
| **45-55** km | | | | |
| Energy integral between 10 and 2 km. $[\mathrm{m^2 s^{-2}}]$ | $36 \pm 25.6$ | $11 \pm 5.8$ | $8.4 \pm 5.5$ | $2.4 \pm 0.8$ |
| slope $[m > m_*]$ | $-2.5 \pm 0.2$ | $-2.4 \pm 0.5$ | $-2.6 \pm 0.04$ | $-2.5 \pm 0.04$ |
| **35-60** km | | | | |
| Energy integral between 10 and 2 km. $[\mathrm{m^2 s^{-2}}]$ | $31 \pm 20.7$ | $8.5 \pm 2.8$ | $5.4 \pm 2.4$ | $2.3 \pm 0.9$ |
| slope $[m > m_*]$ | $-2.06 \pm 0.61$ | $-1.24 \pm 0.51$ | $-2.68 \pm 0.33$ | $-2.72 \pm 0.39$ |

**Table 3.** Table of total potential and kinetic energies per unit mass estimated from integrating wavelengths between 10 and 2 km, and the spectral slopes. The uncertainties are defined as in Tab. 2.

values $\sim 3$ near $m \sim 1/10\,\mathrm{cycle/km}$, which indicates more amplification of longer wavelength waves as altitude increases. This altitude-dependent increase in energy densities aligns with theoretical expectations of wave amplification processes and decreasing atmospheric stability $N$ at higher altitudes. Seasonal variations also remain pronounced, preserving appreciable differences with enhanced GW activity in winter compared to summer, with a similar winter-to-summer ratio as observed at lower altitudes. This ratio behaviour is manifested as a nearly constant value of 2.5 for $E_{pot}$, which is similar to the winter-to-summer ratio between 4–10 h in Fig. 6b. The winter-to-summer ratio for $E_{kin}$, however, increases with increasing wavenumber and peaks at $m \sim 1/3\,\mathrm{cycle/km}$ with a ratio of 4 and then decreases to approach unity as summer amplitudes approach those of winter at high wavenumbers.

To place our observed vertical wavenumber spectra in context, we compare both their slope and amplitude to the predictions of the LIT model defined in Eq. 3. We fit the same bending power-law function previously described in Sec. 3.2 to determine slopes of the mean spectra from different layers, see Tab. 3. For $E_{pot}$ both seasons gravitate toward the canonical $m^{-3}$ predicted by all saturation theories for $m \gtrsim m_*$, but with a season-dependent amplitude offset: in the 35-45 km layer (Fig. 9a) the winter mean is only $\sim 2.5$ times below the LIT curve (on average between 4-1 km), whereas the summer mean is down by a factor of $\sim 6.4$. This offset indicates either reduced source strength or—more plausibly in summer—enhanced pre-saturation filtering in the lower stratosphere that removes part of the low-$m$ tail before it can saturate (Eckermann, 1995). The same filtering signature is visible in the observed-frequency spectra in Fig. 5 and persistently across all altitudes in Fig. 6. In the upper altitude layer (45-55 km), the winter $E_{pot}$ grows greater than the LIT (for $m_* = 1/4\,\mathrm{cycle/km}$), whereas the summer values remain lower, so that both seasons converge toward the theoretical curve at larger scales (lower $m$). This mirrors the quasi-constant winter $\omega$-slope and the marked altitude-dependent flattening of the summer slope reported in Tab. 2 and Sec. 4.1.2. In other words, where the $m$-spectra indicate early saturation (winter) at high wavenumbers, the $\omega$-slope stays steep with height; where the $m$-spectra reveal persistent filtering (summer), the $\omega$-slope flattens as altitude increases and the filtered band is progressively unsaturated (Tsuda et al., 1991).

The $E_{kin}$ spectra show the complementary picture: their high-$m$ (1–3 km) slopes are appreciably shallower than the LIT value of $-3$ (especially in summer), and their amplitudes continue to exceed the LIT saturation curve at these short wavelengths,

even in summer. These seasonal slopes are consistent with radiosonde spectra by Huang et al. (2018). The evident excess kinetic energy at large $m$ explains the flattening of the slope and can be attributed to a broad mixture of near-inertial waves whose polarisation favours kinetic over potential energy. The same waves produce the enhanced $E_{kin}$ bump just above $\sim f$ in the $\omega$-domain, which are not accounted for in the LIT model given in Eq. 3 as discussed earlier. Another interpretation is given by Yoshiki and Sato (2000), who attributed slopes of polar $m$-spectra being flatter than at mid-latitudes to enhanced large $m$ energy in the polar stratosphere, possibly due to secondary GWs from the polar night jet or reduced damping. Just as with the inertial peak in $\omega$-spectra, the bending in the shape of $m$-spectra is more clearly characterized in $E_{kin}$ than in $E_{pot}$ because the slope of the low wavenumber region in $E_{kin}$ is much flatter than in the high-wavenumber regime. However, the mean integrated $E_{pot}$ and $E_{kin}$ between 10-2 km exceed the energy values integrated between 12.8–4.1 h (in the corresponding altitude range) for both seasons, suggesting that part of the energy in this wavenumber region may originate outside this temporal window (e.g., from Doppler shifted GWs with $\omega < f$); see Tab. 3 and Tab. 2.

As altitude increases, the spread between individual $m$-spectra widens markedly, but in a season-dependent fashion. The SD of the winter $E_{pot}$ integrals in the 10-2 km band more than doubles from $1.8\,\mathrm{m^2s^{-2}}$ in 35-45 km to $5.5\,\mathrm{m^2s^{-2}}$ in 45-55 km, while the corresponding summer SD grows from $0.5$ to $0.8\,\mathrm{m^2s^{-2}}$ (Tab. 3). This enhanced winter variability is concentrated at low wavenumbers ($m < 1/3\,\mathrm{cycle/km}$), where the spectrum is most sensitive to intermittency in GW sources such as jets, fronts and topography, in agreement with the source-controlled variance in the unsaturated GW regime (Weinstock, 1990). In contrast, the summer spectrum—dominated by a narrowband GW population rather than by sporadic strong sources—retains a substantially smaller SD. Hence the vertical growth of interseasonal variability is most conspicuous in the $m$-spectra—where it directly mirrors source intermittency and upward amplification—while in the $\omega$-spectra it is partially masked by seasonal differences in Doppler shifting and critical-level attenuation. Although the SD provides an estimate of the interseasonal variability of energy from spectral integrals, the sample size should be taken into account when interpreting these values. This is simply because nearly twice as many soundings were recorded in summer, albeit with a shorter sounding length on average.

## 4.4  $E_{kin}$ to $E_{pot}$ ratio

Comparisons between spectra of $E_{kin}$ and $E_{pot}$ presented in the results section (Sec. 4.1 and Sec. 4.2) reveal that the two forms of energy do not always evolve identically across observed frequencies $\omega$ or wavenumbers $m$, even within the same season. Therefore, in this section, we discuss how the total GW energy is partitioned between $E_{kin}$ and $E_{pot}$ and we use the kinetic to potential energy ratio $R(\omega,m) = E_{kin}(\omega,m)/E_{pot}(\omega,m)$ as a diagnostic of wave character and the underlying dynamics (VanZandt, 1985).

### 4.4.1  Observed frequency spectra

Although both energy density spectra exhibit similar overall shapes (see Fig. 5), their relative magnitudes differ at long periods, reflecting changes in partitioning of the total GW energy depending on the frequency. Both seasonal spectra reveal that at observed periods between 6 and 12 hours in Fig. 5a, $E_{kin}$ grows steeper than $E_{pot}$, suggesting low-frequency enhancement

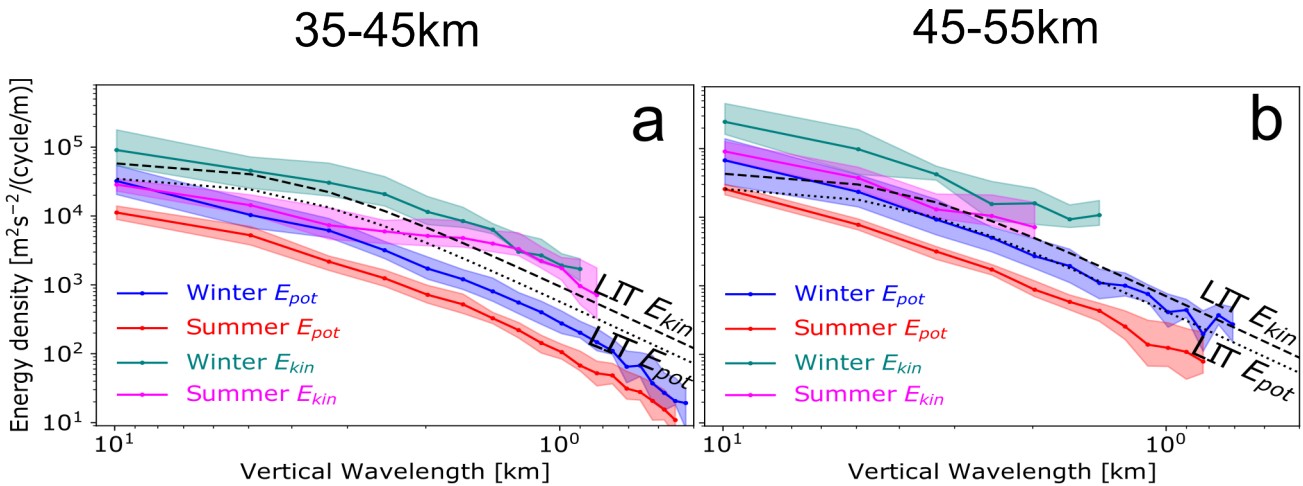

**Figure 9.** Mean vertical spectra, showing the seasonal and altitudinal variability of both $E_{kin}$ and $E_{pot}$. The values of $N^2$ used in the (dashed and dotted) LIT model lines are $46 \times 10^{-5}\,\mathrm{s}^{-2}$ for the layer 35-45 km (a), and $36 \times 10^{-5}\,\mathrm{s}^{-2}$ for the layer 45-55 km (b).

of horizontal-velocity variance near that range. This enhancement was attributed to inertia GW signatures in Sec. 4.1.1. At smaller periods (<6 h) in Fig. 5a, $E_{kin}$ and $E_{pot}$ curves run quasi-parallel, differing mainly by an offset. To understand how the total energy is partitioned along altitude and season, we show $R(\omega)$ from mean observed frequency spectra in Fig. 10, across 5 km layers between 35 km and 60 km. The results clearly demonstrate a dependency of $R(\omega)$ on the season. In summer, with less variable $E_{kin}$, $R(\omega)$ can be described by a rapidly decaying function at low frequencies (<1/6 cycle/h), becoming constant at higher frequencies in Fig. 10b. Summer spectra also show some kind of altitudinal variability of $R(\omega)$ without a direct monotonic relationship; in particular, $R(\omega)$ does not consistently increase or remain constant with altitude. Instead, for observed periods between 7-15 h, $R(\omega)$ decreases as altitude increases. In contrast, winter spectra are broadband and far more random and $R(\omega)$ fluctuates between 6 and 2 with an average of $\approx 4$ at most frequencies, see Fig. 10a. In fact, $R(\omega)$ in winter exhibits maxima at high frequencies (>1/4 cycle/h) which are equal to or higher than $R(\omega)$ at low frequencies. This random altitudinal variation of $R(\omega)$ in winter can be attributed to different Doppler shifting effects at different heights (Scheffler and Liu, 1986).

Earlier observational studies revealed a wide spread in the estimated $R(\omega)$ of GWs. High-latitude case studies—most notably the ALOMAR near-inertial wave analysed by Baumgarten et al. (2015) and climatology of Hildebrand et al. (2017)—showed strong average kinetic energy dominance with $R(\omega) \approx 5 - 10$. In contrast, mid-latitude and tropical climatologies (Vincent et al., 1997; Placke et al., 2013; Ratynski et al., 2025) cluster near the linear-instability theory (Smith et al., 1987) value of $1.5 - 2.5$, and quasi-Lagrangian balloon spectra in the polar lower stratosphere (Podglajen et al., 2020) show a frequency-dependent transition from $\sim 10$ at near-inertial periods to $\sim 1$ at higher frequencies. Radiosonde analyses over South Georgia (Moffat-Griffin et al., 2017) even reported a case where $E_{pot}$ dominates $E_{kin}$ in winter which was attributed to a high intrinsic-frequency orographic wave, demonstrating that latitude, altitude and intrinsic frequency can drive $R(\omega)$ from less than 1 to larger than 10 across the low and middle atmosphere.

As a theoretical reference for the observed $R(\omega)$ in both seasons (dotted line in Fig. 10), we use the relation

$$\frac{E_{kin}}{E_{pot}} = \frac{\hat{\omega}^2 + f^2}{\hat{\omega}^2 - f^2} \tag{4}$$

derived from linear GW polarisation relations for an ideal monochromatic GW in a non-dissipative, hydrostatic atmosphere with intrinsic frequency $\hat{\omega}$ (Gill, 1982; Geller and Gong, 2010). Note again that the intrinsic frequency $\hat{\omega}$ is distinct from the observed frequency $\omega$ employed throughout the preceding sections of this paper. Consequently, we do not expect our observed

spectra to exactly align with theoretical predictions for intrinsic frequency spectra as Podglajen et al. (2020) did, since the non-zero wind magnitudes at the ALOMAR location certainly filter and Doppler-shift observed spectral amplitudes (Fritts and VanZandt, 1987; Mitchell et al., 1994). This theoretical equation (Eq. 4) indicates that energy of high-intrinsic-frequency GWs should be equally partitioned, but as $\hat{\omega} \to f$, the kinetic energy dominates their potential energy (Vincent et al., 1997) and $R(\omega)$ approaches infinity asymptotically. Figure 10b demonstrates that the observed $R(\omega)$ in summer does not exactly match this

theoretical curve for intrinsic frequency spectra (Eq. 4), yet the discrepancy is consistently more pronounced in the observed $R(\omega)$ in winter across all altitudes, see Fig. 10a.

To explain this seasonal modulation, we note that in summer and in all height layers in Fig. 10b, the observed $R(\omega)$ reaches a maximum peak at an observed period of 15 h or longer, corresponding to a frequency below $f$. The opposite happens in winter in Fig. 10a, where the maximum peak is observed at a random period shorter than 12 h (frequency above $f$). This

seasonal contrast reflects the influence of background wind on the observed frequency distribution of GWs. In summer, the background wind shear is weak between 35-60 km, and the day-to-day variability in stratospheric temperature is smallest (Schöch et al., 2008), favouring the persistent propagation of near-inertial waves with narrow frequency spread and with little Doppler shifting (Nastrom and Eaton, 2006). As a result, waves with near-inertial intrinsic frequencies which have high $R(\omega)$ dominate the values at low observed frequencies. Conversely, in winter, there is likely a smaller population of near-inertial

GWs, which are significantly Doppler shifted by stronger winds (Guest et al., 2000; Nastrom and Eaton, 2006), continually redistributing their energy (characterized by high $R(\omega)$) over a broader range of observed frequencies. Consequently, winter $R(\omega)$ show random pronounced peaks across the spectrum in Fig. 10a and show more significant disagreement with theoretical predictions for intrinsic frequency spectra. See Strelnikova et al. (2021) for a detailed picture of wind changes with height at ALOMAR and its interseasonal variability in winter and summer.


### 4.4.2  Vertical wavenumber spectra

Analysis of the vertical wavenumber spectra further complements the analysis along observed frequencies by revealing how $E_{kin}$ relates to $E_{pot}$ without having to account for Doppler shifting effects. Hence, we show the spectra of vertical profiles between 35-60 km in Fig. 11a and their corresponding ratio $R(m)$ in Fig. 11b. Spectra of these 25 km long vertical pro-

files allow us to estimate $R(m)$ at longer wavelengths (up to 25 km) with a higher wavenumber resolution than the 10 km profiles in Fig. 9. In addition, the ratio $R(m)$ estimated within the shorter 10 km layers (35–45 and 45–55 km) show quite

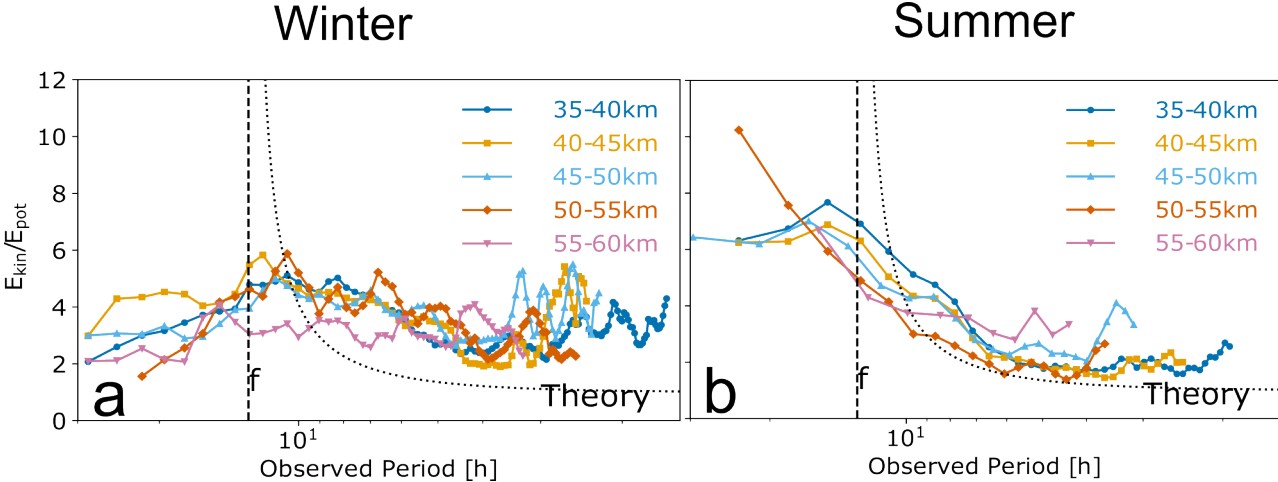

**Figure 10.** Altitudinal variability of the kinetic-to-potential energy-density ratio $R(\omega)$ as a function of observed period for winter (**a**) and summer (**b**) seasons. Curves show the $R(\omega)$ from mean spectra computed in 5 km layers from 35 to 60 km; the black dotted line represents the theoretical relation given by Eq. 4 after (Gill, 1982; Geller and Gong, 2010).

identical $R(m)$ behaviour to the 35–60 km layer. Figure 11b shows that winter $R(m)$ is fairly constant at low wavenumbers ($m < 1/8\,\mathrm{cycle/km}$) where it ranges between 3.5 and 3, increasing as wavenumber increases, reaching values close to 10.8 at wavenumbers near $1/1.5\,\mathrm{cycle/km}$. Summer $R(m)$ increases more substantially with increasing wavenumber (at wavelengths

smaller than 8 km) and reaches $\sim 19.9$ near $1\,\mathrm{cycle/km}$. This increase in both seasons can be characterized by fitting a simple power-law function for $R(m)$ in that range, where slopes of $0.41 \pm 0.08$ and $1.07 \pm 0.07$ are estimated for winter and summer ratios, respectively. At the highest few wavenumbers, however, $R(m)$ seems to fluctuate around fairly constant values in both seasons. While this overall behaviour of $R(m)$ is similar in both seasons across most of the resolved wavenumber range, the $R(m)$ values nonetheless indicate a clear seasonal modulation that can only be explained by differences in the underlying

populations of GWs in each season.

    Comparable *in-situ* evidence for elevated $R(m)$ at high vertical wavenumbers already exists in the literature. For example, seasonal vertical wavenumber spectra from Huang et al. (2018) near the Arctic showed a very similar pattern of increasing $R(m)$ with increasing vertical wavenumbers in the same wavelength range of 1-8 km. At smaller wavelengths (<1 km), however, their spectra apparently showed a fairly constant $R(m)$. In the wavelength range of 0.1-5 km, Nastrom et al. (1997) showed

that the $R(m)$ increases towards a constant ratio of 5 in the stratosphere over mid-latitudes. They hypothesized that it could be matched by a model spectrum which contains a pronounced enhancement of wave energy ($\delta$ function) near the inertial frequency $f$. Likewise, de la Torre et al. (1999) near 32°S, demonstrated values of $R(m)$ which were substantially larger than those predicted by a separable wavenumber-frequency model and increasing with increasing wavenumber. They argued that the high $R(m)$ values reflect a population of near-inertial GWs excited by orography, again implying a kinetic energy surplus

at $\hat{\omega} \sim f$.

    These independent findings strongly suggest that the steep rise of $R(m)$ at short vertical wavelengths (as seen at 1-3 km in

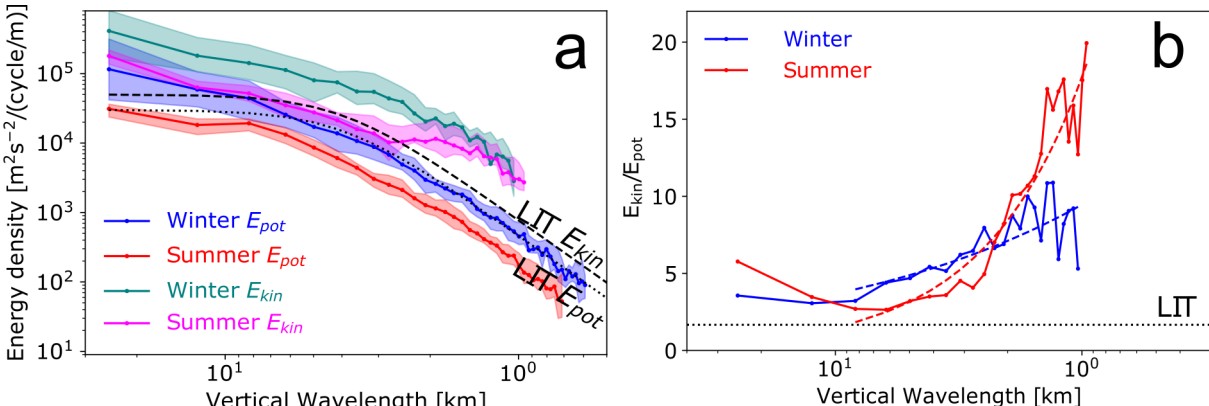

**Figure 11.** (a) Mean vertical spectra showing the seasonal variability of both $E_{kin}$ and $E_{pot}$ for the height range 35-60 km. The values of $N^2$ used in the LIT model lines are $39 \times 10^{-5}\,\mathrm{s}^{-2}$. (b) The corresponding ratio of kinetic to potential energy densities $R(m)$ at different vertical wavelengths.

Fig. 11b) is produced by a distinct population of GWs whose intrinsic frequencies $\hat{\omega}$ lie very close to the Coriolis frequency $f$. This implication is consistent with the stratospheric results from Thompson (1978) at 38°S which showed that the kinetic energy concentrated near the local inertial frequency is associated with a vertical wavelength between 1 and 3 km. Moreover,
as noted by Conway et al. (2019), inertia-GWs typically have $\hat{\omega} \sim f$, long horizontal wavelengths, and vertical wavelengths $\sim 1$ km. In order to test whether these high $R(m)$ values in our results are due to near-inertial GWs, we first need to isolate waves in the high-wavenumber range where there is a $E_{kin}$ surplus, and determine to what part of the frequency spectrum they belong. This is achieved by passing the perturbations through a sixth-order Butterworth band-pass filter that retains only the 1-3 km band. We then explore how the observed frequency spectra of the filtered perturbations differ from the unfiltered per-
turbations, see Fig. 12 for $\omega$-spectra between 35-40 km. An obvious difference emerges: we see that the filtered spectra exhibit a sharp energy drop from the unfiltered spectra at all frequencies, yet the overall spectral shape and extent stay nearly the same. Furthermore, the spectral slopes remain close to the canonical -5/3 value (dashed reference lines in Fig. 12), indicating it is not a property of long-wavelength motions only, but persists even when only the 1-3 km band is retained. This means that there are contributions from all (resolvable) observed frequencies to this 1-3 km band, but with differing relative amplitudes. So a
natural question arises: is there evidence suggesting which intrinsic frequencies correspond to these observed ones?

To address this question, we assess the variation in $E_{kin}$ and $E_{pot}$ after filtering by examining the following from Fig. 12:

- Differences in attenuation patterns between summer and winter,

- The frequency dependence of the attenuation (i.e., whether it is uniform or concentrated near certain frequencies),

- The relative attenuation of $E_{kin}$ versus $E_{pot}$.

The filtered spectra indeed reveal a noticeable seasonal contrast in both $E_{kin}$ and $E_{pot}$. In summer, as $\omega$ decreases to frequencies as low as $1/20\,\mathrm{cycle/h}$, the more the spectra of the filtered perturbations closely match the spectra of the unfiltered

perturbations. At higher frequencies ($>1/8\,\mathrm{cycle/h}$), attenuation is much stronger. While this might be true overall, the filtered spectra exhibit a more pronounced attenuation of $E_{pot}$ than $E_{kin}$, with the $R(\omega)$ increasing to $\sim 16$ just below $f$. Figure 12a shows that the $E_{kin}$ peak just above the inertial period (around 12.8 h) survives the filtering almost intact. In contrast, as the energy at higher frequencies drops by about an order of magnitude, the $R(\omega)$ remains relatively constant at a value of approximately 5. In fact, the filtered summer spectra reach levels as high as (or even higher) the filtered winter spectra at low frequencies. Hence, in summer, the high $R(m)$ obtained in the 1-3 km band (seen in Fig. 11b) are dominated primarily by low-observed-frequency waves with similar high $R(\omega)$ values.

The filtered winter spectra draw the opposite picture, i.e. the filtered spectra diverge substantially away from the unfiltered spectra as $\omega$ decreases towards frequencies lower than $1/6\,\mathrm{cycle/h}$ for $E_{kin}$ and $1/8\,\mathrm{cycle/h}$ for $E_{pot}$. At higher frequencies, both filtered $E_{kin}$ and $E_{pot}$ exhibited smaller but uniform attenuation offset. This is in contrast to the summer spectra, where the low-frequency part of the spectrum is the least affected part by filtering. Nonetheless, the winter filtered $E_{pot}$ still lost more of its variance than $E_{kin}$ between $1/f$ and 8 h, compare Fig. 12b and Fig. 12a. This difference at low frequencies makes $R(\omega)$ reach a value of 10, which is also about double the ratio in the unfiltered spectra in that range. This suggests that winter $R(m)$ ratio at vertical wavelengths between 1-3 km, are comprised of contributions from a broader observed frequency range, rather than being as much dominated by low-observed-frequency waves as in summer.

Combining these frequency-domain results with

1. the seasonal modulation of $R(m)$ (in Fig. 11b),

2. the better qualitative agreement of summer $R(\omega)$ with Eq. 4 (for intrinsic frequencies) than winter, which could be primarily due to different populations of near-inertial GWs and stronger Doppler shifting in winter (Fig. 10),

suggests the following interpretation. The order-of-magnitude $R(m)$ values observed between 1-3 km can be attributed to an enhancement of the $E_{kin}$ relative to $E_{pot}$ at near-inertial intrinsic frequencies. In summer, these waves are only slightly shifted to nearby frequencies, whereas in winter they are likely rarer and substantially shifted to a broader distribution at higher and lower frequencies. Together, the filtered spectra and frequency and wavenumber ratio scalings converge on a coherent interpretation: a seasonally modulated population of near-inertial, short-wavelength GWs dominates the upper-stratospheric energy budget at high latitudes by channelling a disproportionate share of energy into kinetic form.

**Implications for spectral separability:**

These findings carry significant implications for the underlying structure of the GW energy spectrum. Specifically, the strong and systematic variation of the kinetic-to-potential energy ratio with both observed frequency and vertical wavenumber, as well as the frequency-dependent response to vertical bandpass filtering, indicate a clear breakdown of spectral separability in the $(m,\omega)$ domain. If the joint energy spectrum $E(m,\omega)$ were separable, i.e., expressible as the product of independent functions of vertical wavenumber and frequency, the ratio $E_{kin}/E_{pot}$ would be invariant under filtering in either domain. However, our observations show that filtering the perturbations in vertical wavenumber space induces substantial, frequency-dependent changes in energy magnitudes and in the energy ratio, demonstrating cross-dependencies between vertical and temporal wave characteristics. These results thus align more closely with the predictions of a frequency-dependent theoretical ratio (Geller and

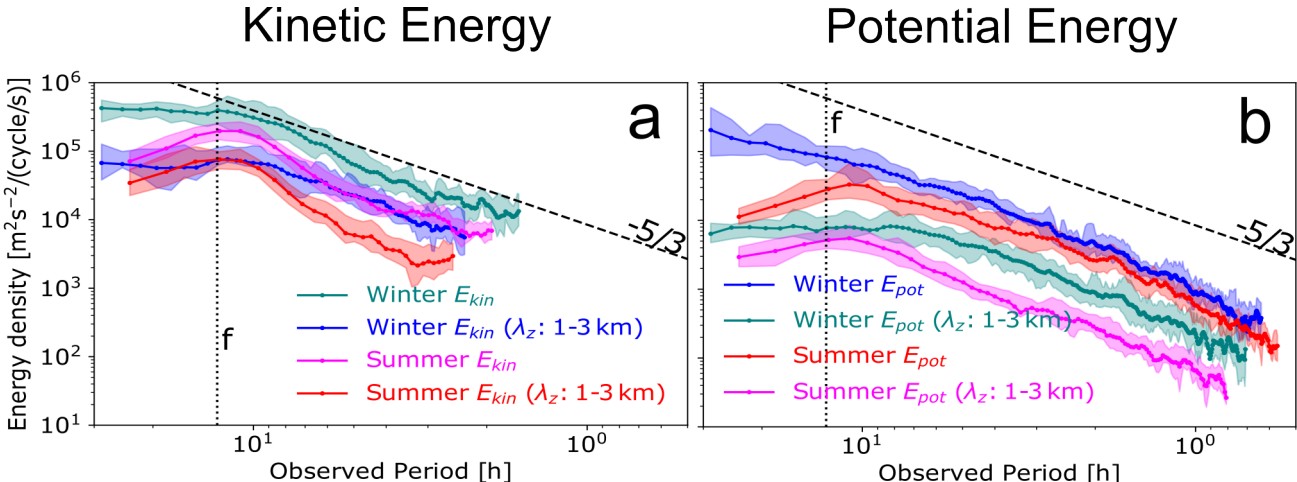

**Figure 12.** Mean observed frequency spectra, showing the seasonal variability of both $E_{kin}$ (a) and $E_{pot}$ (b) after filtering the perturbations using a 6-th order bandpass Butterworth filter with a cut-off of 1-3 km.

Gong, 2010) and contradict the assumption of a small constant-ratio, which is often taken to imply separability as is typically assumed in models based on the linear instability theory, see Eq. 3 (Smith et al., 1987; Nastrom et al., 1997). Importantly, other theoretical frameworks also predict or assume non-separability of the intrinsic spectrum, such as the diffusive filtering theory and the saturated cascade theory (Gardner, 1996). Fritts and Alexander (2003) also pointed out that separability is assumed only 740 for convenience and is not "fundamental to any saturation theory". Furthermore, as discussed by Gardner et al. (1993), even if the intrinsic spectrum were separable, the observed spectrum is likely not separable (particularly at high wavenumbers) due to Doppler shifting and filtering effects. Comparable evidence against the spectral separability has been reported in lidar and radiosonde studies. For instance, Li et al. (2021) showed that in the mesopause region over the Andes, the spectra of horizontal winds are separable only at high frequencies. Similarly, the vertical wind spectrum by Gardner et al. (1995) demonstrated 745 inconsistency with the concept of a separable GW spectrum. Pramitha et al. (2021) also reported that the observed vertical wavenumber spectra of GWs in the lower stratosphere over the tropics do not conform to the assumptions of separability. Together with supporting evidence from prior observational studies, our results reinforce the conclusion that GW energy spectra in the upper stratosphere cannot be adequately described by separable or constant-energy-ratio models.

**5   Summary and conclusions**

This study provides the first multi-year (2017 to 2023) climatology of gravity wave (GW) kinetic and potential energy spectra derived from simultaneous temperature and horizontal wind measurements from the daylight-capable Doppler Rayleigh lidar at ALOMAR (69°N, 16°E). 61 soundings, each longer than 12 h, were analysed between 35-60 km altitude, from which observed-frequency ($\omega$) and vertical-wavenumber ($m$) spectra of kinetic and potential energy densities ($E_{kin}$ and $E_{pot}$) were

derived. Our unique data enabled a direct, altitude-resolved assessment of the winter-summer variability of kinetic-to-potential

energy in both domains ($\omega$ and $m$), $\omega$-spectral slopes and a comparison with a linear-instability theory (LIT) model.

Our key findings include:

**Seasonal contrast of energy spectra:**

Throughout the mid-stratosphere to the lower-mesosphere, winter spectra carry roughly 3–7 times more potential energy at

intermediate and long periods and wavelengths and 2–4 times more kinetic energy than their summer counterparts at periods

smaller than $8\,\mathrm{h}$ and wavelengths larger than $3\,\mathrm{km}$, reflecting stronger GW activity and weaker critical-level filtering at the

lower altitudes in winter. At shorter vertical wavelengths ($\lambda_z \lesssim 3\,\mathrm{km}$) and periods close to 8–10 h the seasonal contrast shrinks

for $E_{kin}$, so that summer $E_{kin}$ closely matches winter values, whereas summer $E_{pot}$ remains smaller across the resolvable

$m$-range and only approaches winter values at very short periods ($< 2\,\mathrm{h}$).

**Departure from LIT model:**

The observed average $E_{pot}(m)$ in summer lies a factor 6.4 below the canonical LIT curve, while winter $E_{pot}$ is only 2.5 times

smaller in the stratosphere and slightly greater in the mesosphere. In contrast, $E_{kin}(m)$ exceeds the LIT limit at $\lambda_z < 3\,\mathrm{km}$ by

factors of 3.2 (winter) and 1.5 (summer). A bending power-law fit shows that the high-wavenumber slope of $E_{kin}$ is markedly

shallower than the canonical $-3$, indicating that the LIT under-predicts velocity variance in the near-inertial, short-wavelength

regime.

**Altitudinal evolution:**

Layer-to-layer amplification of total energy (both $E_{pot}$ and $E_{kin}$) follows the expected $\exp(z/H)$ scaling with $H \approx 7$–$10\,\mathrm{km}$,

yet the kinetic amplification is systematically larger than the potential one. Above $50\,\mathrm{km}$ the pronounced inertial peak in

$E_{kin}(\omega)$ broadens and weakens, while high-frequency components grow more rapidly, flattening the high-frequency band

slope in $E_{pot}$ in both seasons. This gradual flattening with height causes the observed broken power-law spectra (reported here

for the first time) to merge into a single spectrum. Additionally, there is no significant altitudinal variation of the winter-to-

summer amplitude ratio in all $5\,\mathrm{km}$ layers in the 35–60 km range.

**Kinetic-to-potential partition and near-inertial dominance:**

The ratio $R(\omega, m) = E_{kin}/E_{pot}$ remains relatively small ($\approx 3$) at $m < 0.1\,\mathrm{cycle\,km^{-1}}$ but rises as a power law to $R \sim 10$ and

$R \sim 20$ at wavelengths between 1–3 km in winter and summer, respectively. In the frequency domain, $R(\omega)$ peaks at periods

just below the local inertial period in summer and shifts to more randomly shorter periods in winter, consistent with Doppler

shifting by the stronger wind in winter. Band-pass filtering of the 1–3 km vertical-wavelength band confirms that a seasonally

modulated population of near-inertial, short-wavelength waves channels a disproportionate share of the total energy into kinetic

form, dominating the upper-stratospheric energy budget.

**Spectral separability breaks down:**

Because $R(\omega, m)$ varies systematically in both domains, the joint spectrum $E(\omega, m)$ cannot be factorised into independent

functions of $\omega$ and $m$. Vertical band-pass filtering induces frequency-dependent changes in $R$, providing direct observational

evidence that the observed separable-spectrum assumption used in many GW parametrisations is likely untenable.

**Implications:**

(i) LIT models provide only a lower bound on kinetic energy at high $m$; (ii) the kinetic-to-potential partition is strongly frequency- and wavenumber-dependent; and (iii) seasonally varying near-inertial waves dominate short-vertical-scale dynamics at high latitudes. These findings challenge parametrisations that assume constant $E_{kin}/E_{pot}$ or spectral separability. Further observational research is needed to identify GW sources and better understand propagation and dissipation mechanisms behind observed seasonal and altitudinal variability. Given ongoing climate changes in polar regions, enhanced understanding
of GW dynamics at high latitudes is critical for accurate climate modelling and forecasting.

*Data availability.*  The data files to reproduce all figures in this study are accessible on RADAR's link
(DOI after publication: 10.22000/nj61edyby3z3xrhe)

*Author contributions.*  M.M. and G.B. contributed to recording part of the data. M.M. analysed the data and drafted the manuscript. I.S., R.W., and G.B. provided supervision, scientific insight, and edited the manuscript. M.G. provided scientific insight, and edited the manuscript.

*Competing interests.*  The authors declare no competing interests.

*Acknowledgements.*  We gratefully acknowledge all colleagues and staff at the ALOMAR observatory whose dedication to lidar operation, data recording, and measurement support made this study possible. This research is a contribution to the project W1 (Gravity Wave Parameterization for the Atmosphere) of the Collaborative Research Centre TRR 181 "Energy Transfers in Atmosphere and Ocean" funded by the Deutsche Forschungsgemeinschaft (DFG, German Research Foundation) - Projektnummer 274762653 and the Analyzing the Motion of
the Middle Atmosphere Using Nighttime RMR-lidar Observations at the Midlatitude Station Kühlungsborn (AMUN) funded by Deutsche Forschungsgemeinschaft (DFG) - Projektnummer 445400792. I.S. was supported by the Federal Ministry for Economic Affairs and Climate Action on the basis of a decision by the German Bundestag (DLR grant 50OE2301, project DEFINE). G.B. gratefully acknowledges the financial support of the NATO Science for Peace and Security (SPS) Programme under grant G6122.

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
