# Peer review of "Spectral variability of gravity-wave kinetic and potential energy at 69°N: a seven-year lidar study"

_EGUsphere, 2025_

## Author Comment (AC1)

**Response to Reviewer #1**

*We appreciate the comments, questions and references given by both reviewers.*
*We repeat the reviewers' concerns and provide our respective responses in italics. The changes take place in the revised manuscript.*

The work "Spectral variability of gravity-wave kinetic and potential energy at 69∘N: a seven-year lidar study" by Mossad and co-authors constitutes a comprehensive study of a unique dataset. The results are very relevant for this field of research and provide a reference for other studies and observations. The analysis is carefully done, and the presentation of results, the interpretation of measurements and the implications are well written. I have no major comments or questions, and recommend publication after minor corrections which are listed below.

> *Thank you for the nice words and appreciation.*

Minor comments:

Abstract: In the abstract 2700 h of measurements and 100 soundings are mentioned, but the actual data used is less. Only summer and winter data are actually used in the study, e.g. from months Jan-Feb and Jun-Jul-Aug. They amount to 1091 h and 745 h according to Fig. 1 and Table 1. The actual number of used soundings is not given. I recommend to add these numbers to the abstract, and correct the number of soundings in line 742.

> *Done*

line 3: comprising --> comprises

> *Changed*

line 65: add reference https://agupubs.onlinelibrary.wiley.com/doi/10.1029/2023GL104357

> *Added*

line 72: delete "will", delete "ever"

> *Removed*

line 80: uniqueness of the ALOMAR location: also mention that it is situated at the coast and close to the Scandinavian mountain ridge

> *Done*

line 82: "only instrument in the world" change sentence to
include  https://agupubs.onlinelibrary.wiley.com/doi/full/10.1002/2016JD026368 and
https://agupubs.onlinelibrary.wiley.com/doi/full/10.1029/2017JD027386

*Changed to :" Moreover, the daytime capability of the ALOMAR lidar allows continuous measurements of stratospheric and mesospheric GW activity even during polar summer, similar to the Antarctic Fe lidars at the McMurdo Station (Chu et al., 2018; Zhao et al., 2017a) and previously at the Davis Station (Kaifler et al., 2015). However, the ALOMAR Doppler system is unique in that it provides long-term simultaneous temperature and horizontal wind observations using its twin-beam configuration."*

line 97: "measuring horizontal wind velocities for 30 years" is that true? The Fiedler and Baumgarten (2024) reference only gives one example from 2017. Suggest to add "the capability for wind measurements was added in xx" or similar

*It was changed to "The capability for current horizontal wind measurements was reported in 2010 (Baumgarten, 2010), with first wind observations in 2009."*

line 90: Sec. 2.4 --> Sec. 4.4

*Done*

line 105: two 1.8 m telescopes --> two tiltable 1.8 m telescopes

*Done*

line 131: 27.5 h --> 27.6 h

*Done*

line 143: It is not clear what a Reynolds decomposition is. Please explain.

*Done*

line 145: delete "and turbulence". I don't think this can be seen at 5 min resolution.

*Done*

line 147: Is there a difference between removing a 12-h running mean and a sounding-length running mean because the sounding length differs from sounding to sounding?

*We do not subtract a temporal running mean as an estimation of background, we do subtract temporal constant mean at each altitude grid. But in general, yes, subtracting a 12 hour running mean would remove variability on timescales longer than 12 h consistently across all soundings. However, since our data is ground-based, waves which are Doppler-shifted to observed periods longer than 12 hours would be omitted from frequency and vertical wavenumber spectra. Thus, had we used a 12 h running mean as a background, we would not have been able to observe the seasonally modulated Synoptic gap, winter-summer ratio contrast, distinct inertial peak in summer Ekin at frequencies close to and lower than f and their corresponding wavenumber behavior.*

line 150: "the problem of long vertical stripes" I didn't find that in the reference. What kind of problem is that, an instrumental problem?

*From the cited (Zhao 2017a) reference: "then subtracting the altitudinal mean at each time grid for every observational segment. Such altitudinal mean subtraction is to remove the nearly vertical stripes found in some segments of the Rayleigh temperature data, equivalently removing waves with long vertical wavelengths."*

*They arise from photon noise if the backscatter signal is weak, which is amplified by the sensitive inversion procedure used to derive temperature from density. Additional contributions come from binning and background subtraction, which can create long vertically-correlated noise that appears as striping in time-height plots.*

line 258: delete "fine"

> *Done*

line 259/260: 5 min resolution and 1221 individual spectra for winter adds to 102 h. Isn't winter supposed to comprise 745 h?

> *This is about the average vertical wavenumber spectrum of the winter case (30 January-4 February 2018) which was 105 hour long in total, with a resolution of 5 min -> 5 min\*1221 selected vertical profiles /60 min = 101.75 hours. The 745 h are the sum of observed hours from all winter soundings.*

line 260: "is much smoother" Is that also because of the top-down integration?

> *No, this has nothing to do with integration, it is just because of more averaging over time. The minimum sounding length ~12 hours -> has 12 hours /5 min= 144 vertical profiles, thus the average vertical wavenumber spectrum of this sounding corresponds to the mean of those 144 individual vertical wavenumber spectra. While in a 5 km range (e.g. 35-40 km) and 150 m vertical resolution, there are only 34 time series, and thus the average frequency spectrum of this layer in any sounding corresponds to the mean of those 34 individual frequency spectra.*

line 283: what is "p" in the equation "b approx pd approx 1/6"?

> *p is the the slope of intrinsic frequency spectrum, it is defined in the previous section in line 238 and the succeeding line 284. We adjusted this line to make it clearer.*

Fig. 3: I suggest to add the dates in the legend to make clear that those are single cases and not winter and summer averages

> *Done*

line 332: upper stratosphere --> mid stratosphere?

> *Done*

line 345: measurements in (Hertzog.. --> measurements by Hertzog...

*Done*

Table 2: For winter Epot energy a digit is missing for the uncertainty ("1"), in line 425 it is "1.4"

*Adjusted*

line 604: rabidly --> rapidly

*Done*

line 676: "we need to first" --> "we first need to"

*Done*

line 677: delete "do"

*Done*

line 683: change "an artefact of" to "it is not a property of long-wavelength motions only, but.."?

*Done*

line 707-713: this sentence is too long

*Done*

check the use of brackets around citations: at least in l. 63, l. 64, l. 345, l. 350, l. 613, l. 662, l. 667 \citep should be changed to \citet

*Done*

---

## Author Comment (AC2)

**Response to Reviewer #2**

*We appreciate the comments, questions and references given by both reviewers.*
*We repeat the reviewers' concerns and provide our respective responses in italics. The changes take place in the revised manuscript.*

As with Reviewer 1, I find this paper well-written, clear and interesting. The dataset is an interesting one, the analysis is well-done, and the work is clearly presented with only extremely minor errors, nearly all of which are typographical.

    *Thank you for the nice words and appreciation.*

I concur with all the points made by reviewer 1, and hence see no need to duplicate them again. I have the following additional typos, but I would be fine with the paper being accepted pretty much as-is, and definitely if the corrections suggested by Reviewer 1 were made:

L010 showed -> 10, but the sentence is a little tricky to read as a whole so couod be rephrased.

    *We changed the abstract to make it more easily readable.*

L013 budget -> budgets ("or an accurate...")

    *Done*

L020 - contribute significantly

    *Done*

L035 - a clearer definition of m* might be helpful here

    *Done*

L055: specific mention of exact instrument resolution jars - is this number special in some physical way, or just what your instrument measures?

    *Indeed, the numbers themselves are not significant in this sentence, although high resolutions are required to resolve GWs. We adjusted the sentence.*

L063 - don't think these references should be bracketed? Same for the rest of the paragraph

    *Done*

L065: they can also derive non-energy things! Suggest a slight rephrase :-)

*Rephrased*

L083: why is "midnight sun" capitalised?

*Changed*

L126: gain ->gaining, provide -> providing

*Done*

L134: noctilucent (not capitalised)

*Done*

L188: interleaved (again, not capitalised)

*Done*

Intro to section 3 is a  single extremely long paragraph - maybe chop in 2 or 3?

*Done*

L289: expand out "approximately"

*Done*

L492: "conforming" is being used here as a calque of the German term I think - in English it think you're after something like "consistent with previous work"?

*It was adjusted to "showing consistent behaviour within each season".*

L496: "severely" -> "significantly"

*Changed to "substantially"*

L508: "significant" -> "significantly"

*Adjusted to "steeper slopes (which are statistically significant)"*

L584: "small" -> "smaller"

*Done*

L683 "even when only the 1-3km band"

*Done*

L728: I wouldn't capitalise the name of these theories

*Done*